# Membrane fluidity is regulated by the *C. elegans* transmembrane protein FLD-1 and its human homologs TLCD1/2

Mario Ruiz[1†], Rakesh Bodhicharla[1†], Emma Svensk[1†], Ranjan Devkota[1], Kiran Busayavalasa[1], Henrik Palmgren[1,2], Marcus Ståhlman[3], Jan Boren[3], Marc Pilon[1*]

[1]Department of Chemistry and Molecular Biology, University of Gothenburg, Gothenburg, Sweden; [2]Diabetes Bioscience, Cardiovascular, Renal and Metabolism, IMED Biotech Unit, AstraZeneca, Gothenburg, Sweden; [3]Department of Molecular and Clinical Medicine/Wallenberg Laboratory, Institute of Medicine, University of Gothenburg, Gothenburg, Sweden

**Abstract** Dietary fatty acids are the main building blocks for cell membranes in animals, and mechanisms must therefore exist that compensate for dietary variations. We isolated *C. elegans* mutants that improved tolerance to dietary saturated fat in a sensitized genetic background, including eight alleles of the novel gene *fld-1* that encodes a homolog of the human TLCD1 and TLCD2 transmembrane proteins. FLD-1 is localized on plasma membranes and acts by limiting the levels of highly membrane-fluidizing long-chain polyunsaturated fatty acid-containing phospholipids. Human TLCD1/2 also regulate membrane fluidity by limiting the levels of polyunsaturated fatty acid-containing membrane phospholipids. FLD-1 and TLCD1/2 do not regulate the synthesis of long-chain polyunsaturated fatty acids but rather limit their incorporation into phospholipids. We conclude that inhibition of FLD-1 or TLCD1/2 prevents lipotoxicity by allowing increased levels of membrane phospholipids that contain fluidizing long-chain polyunsaturated fatty acids.
**Editorial note:** This article has been through an editorial process in which the authors decide how to respond to the issues raised during peer review. The Reviewing Editor's assessment is that all the issues have been addressed (see decision letter).
DOI: https://doi.org/10.7554/eLife.40686.001

*For correspondence:
marc.pilon@cmb.gu.se

†These authors contributed equally to this work

## Introduction

Membrane homeostasis has been a defining feature of cells since their very origin several billion years ago (*Deamer, 2017*; *Deamer et al., 2002*; *Monnard and Deamer, 2002*). Today we find numerous instances of membrane composition adaptation to environmental conditions such as temperature, pH or pressure across all of life's diversity (*Siliakus et al., 2017*; *Guschina and Harwood, 2006*; *Cossins and Macdonald, 1989*). Further, the specific composition of subcellular membranes is of paramount importance for many cellular processes ranging from vesicular trafficking to organelle homeostasis and receptor signaling (*Holthuis and Menon, 2014*). In particular, recent studies suggest important roles for specific polyunsaturated lipids in membrane deformation (*Pinot et al., 2014*), domain stability (*Leventental et al., 2016*) and membrane fission events (*Manni et al., 2018*). It is therefore not surprising that many disease states are associated with membrane composition defects. Diabetics, for example, have rigid cellular membranes rich in saturated fatty acids (*Borkman et al., 1993*; *Kröger et al., 2015*; *Baba et al., 1979*; *Winocour et al., 1990*) that likely contribute to many aspects of the pathophysiology of this disease such as poor microcirculation and

**eLife digest** The saying "you are what you eat" is probably closest to the truth when talking about fats. This is because the outer edge of our cells, known as the cell membrane, is built mostly from the fats found in our diet. Yet, there is a possible problem with this situation. Based on their chemical make-up, fats can be classed as saturated or unsaturated. Saturated fats, like those in butter, pack closely together and tend to be solid at room temperature. Unsaturated fats, like oils from vegetables and fish, pack together more loosely, making them runny. This means that eating too much saturated fat can cause our cell membranes to become unhealthily rigid, while too much unsaturated fat may make them too fluid.

To avoid these problems, cells can manage the fat content of their membranes, even as the diet fluctuates. Certain proteins alert cells when the membrane is too rigid, and the cells respond by converting saturated fats in the membrane into unsaturated fats. Previous research has found a few of these proteins in humans and nematode worms, but it was not clear what would happen if these sensors were not present or not working as they should. Could other proteins monitor the membrane too?

To answer this question, Ruiz, Bodhicharla, Svensk et al. took nematode worms that could not make the known sensor proteins and used genetic methods that stop them producing various other proteins too. The mutant worms were then fed a diet rich in saturated fats, and the effect on the fats in the cell membrane was measured. From these experiments, new proteins were found that regulate the composition of fats in cell membranes. In the worms, it turns out that a protein called FLD-1 normally limits the amount of unsaturated fats in the membrane. Stopping the worms from producing this protein made the membrane less rigid. The same applied in human cells with proteins called TLCD1 and TLCD2. More unsaturated fats were incorporated when these two proteins were removed, restoring the membranes to a healthy state despite the cells being grown in the presence of excess saturated fat. Together, these findings suggest that targeting these proteins could potentially lead to new treatments for diseases like diabetes, where cell membranes are too rigid.

DOI: https://doi.org/10.7554/eLife.40686.002

impaired insulin signaling (*Gianfrancesco et al., 2018*; *Pilon, 2016*; *Samuel and Shulman, 2012*), and may be a useful diagnostic criterion (*Cordelli et al., 2018*). Membrane properties also influence important aspects of cancer cells (*Nicolson, 2015*) and changes in membrane composition correlate with transformation, for example in breast cancer cells (*Hilvo et al., 2011*; *Dória et al., 2012*). Indeed, the precise composition of membrane phospholipids influences the activity of many membrane proteins and must often be balanced against possible trade-offs. For example, the levels of omega-3 long-chain polyunsaturated fatty acids (LCPUFAs; i.e. fatty acids of 18 or more carbons and at least two double bonds) are critical for the activity of TRP channels that regulate blood pressure in mammals (*Caires et al., 2017*) or mechanosensation in the nematode *C. elegans* (*Vásquez et al., 2014*). Conversely, these same LCPUFAs are highly prone to peroxidation via Fenton and Haber-Weiss reactions, and therefore represent a liability for the cell (*Gaschler and Stockwell, 2017*; *Ayala et al., 2014*; *Kelley et al., 2014*).

Given its central and far-reaching importance, it is surprising that so little is known about the molecular mechanisms that regulate membrane composition and fluidity, a term used throughout this article as a proxy for membrane properties that include fluidity, phase behavior, thickness, curvature, intrinsic curvature and lateral pressure profile (*Radanović et al., 2018*; *Mouritsen, 2011*). New or improved methods, such as lipidomic analysis of membrane composition (*Löfgren et al., 2016*; *Ivanova et al., 2009*; *Ståhlman et al., 2009*; *Triebl et al., 2017*; *Jurowski et al., 2017*), fluorescence recovery after photobleaching (FRAP) and other methods to measure membrane fluidity *in vivo* (*Maekawa and Fairn, 2014*; *De Los Santos et al., 2015*; *Devkota and Pilon, 2018*), molecular modelling of membranes (*Berkowitz, 2009*; *Lin et al., 2016*; *de Vries et al., 2005*) and powerful genetic approaches in bacteria, yeast and animals have recently led to the identification of regulators of membrane homeostasis, and elucidation of their mechanism of action. The prototypical fluidity regulator is the prokaryotic DesK protein, which acts as a fluidity sensor that changes

conformation depending on membrane thickness, hence switching between its phosphatase and kinase activities and thus regulating its target, a fatty acid desaturase that promotes membrane fluidity when activated (*Saita and de Mendoza, 2015*; *Abriata et al., 2017*; *Cybulski et al., 2015*; *Inda et al., 2014*). In yeast, the protein Mga2 senses membrane saturation in the endoplasmic reticulum (ER), the site of phospholipid synthesis, via its transmembrane helix, thereby regulating the OLE pathway that promotes fatty acid desaturation (*Covino et al., 2016*; *Ballweg and Ernst, 2017*). Similarly IRE1, which is also located in the ER, maintains membrane homeostasis by responding to changes in lipid order (*Radanović et al., 2018*; *Halbleib et al., 2017*).

Maintenance of membrane homeostasis in animals is especially challenging since their structural fatty acids (FAs) are primarily obtained from a highly variable diet. For example, the nematode *C. elegans* daily replaces nearly 80% of its membrane phospholipids using mostly dietary fatty acids as building blocks (*Dancy et al., 2015*). Given the great variation in dietary composition and the narrow membrane composition range optimal for cellular functions, it is evident that robust regulatory mechanisms must exist that adjust FA composition and compensate for dietary variation. We have previously leveraged the power of forward genetics in the nematode *C. elegans* to demonstrate that the proteins PAQR-2 and IGLR-2 act as sensors in the plasma membrane that are essential to promote fatty acid desaturation and restore fluidity during cold adaptation or when fed diets rich in membrane-rigidifying saturated fatty acids (SFAs) (*Devkota et al., 2017*; *Svensk et al., 2016a*; *Svensk et al., 2013*; *Svensson et al., 2011*). Similarly, the human AdipoR1/2 that are homologs of the *C. elegans* PAQR-2, also regulate membrane fluidity in human cells (*Bodhicharla et al., 2018*; *Devkota et al., 2017*). Given the importance of membrane homeostasis, we reasoned that other regulatory pathways likely exist to ensure its robust maintenance.

## Results

### *fld-1* mutations suppress membrane fluidity defects

We used forward genetics in *C. elegans* to isolate novel mutants that suppress the lethality phenotypes of strains with membrane homeostasis defects. Specifically, we performed screens in the *paqr-2(tm3410)* and *iglr-2(et34)* mutant backgrounds because these mutants are unable to maintain membrane fluidity when cultivated at low temperatures, which rigidifies membranes, or in the presence of glucose, which is readily converted to SFAs by the dietary *E. coli* (*Devkota et al., 2017*; *Svensk et al., 2016a*). *paqr-2(tm3410) mdt-15(et14)* double mutants were also screened in an effort to identify novel *paqr-2* suppressors that act independently from the desaturases activated by the gain-of-function (*gof*) *mdt-15(et14)* allele (*Svensk et al., 2013*). In all, over 100 000 mutagenized haploid genomes were screened, leading to the isolation of 15 independent mutants that suppress the glucose intolerance of the sensitized strains. The mutants were outcrossed several times to remove unwanted mutations and their genomes sequenced, which revealed eight novel alleles of *fld-1* (membrane fluidity homeostasis-1; *Figure 1A* and *Figure 1—figure supplements 1–2*), a previously uncharacterized gene; the other mutations affect different genes and remain uncharacterized. Introducing a wild-type *fld-1* transgene abolished the suppression, demonstrating that the novel *fld-1* alleles are loss-of-function mutations (*Figure 1B–D*). This was further confirmed using the loss-of-function (*lof*) *fld-1(gk653147)* mutant obtained independently by the *C. elegans* one-million-mutation project (*Thompson et al., 2013*) and that also suppressed the glucose intolerance of the *paqr-2* mutant (*Figure 1—figure supplement 3A*). Western blot assays using an antibody against a C-terminal peptide show that the FLD-1 protein is absent from the *fld-1(et48)* mutant (*Figure 1—figure supplement 3B*). The fact that 8 of the 15 novel mutations that suppress SFA toxicity in sensitized *C. elegans* backgrounds were alleles of *fld-1* suggest that mutating this gene is the easiest genetic mean to prevent this toxicity.

### *fld-1* is ubiquitously expressed in plasma membranes

*fld-1* encodes a multiple-transmembrane domain protein homologous to the human TLCD1 and TLCD2 proteins (*Figure 1E*) that belong to a protein family characterized by the presence of a TLC domain and that includes the ER-localized ceramide synthases as well as translocation associated membrane proteins involved in membrane protein synthesis in the rough ER (*Pewzner-Jung et al., 2006*; *Winter and Ponting, 2002*). The TLC domain was named after three members of this large

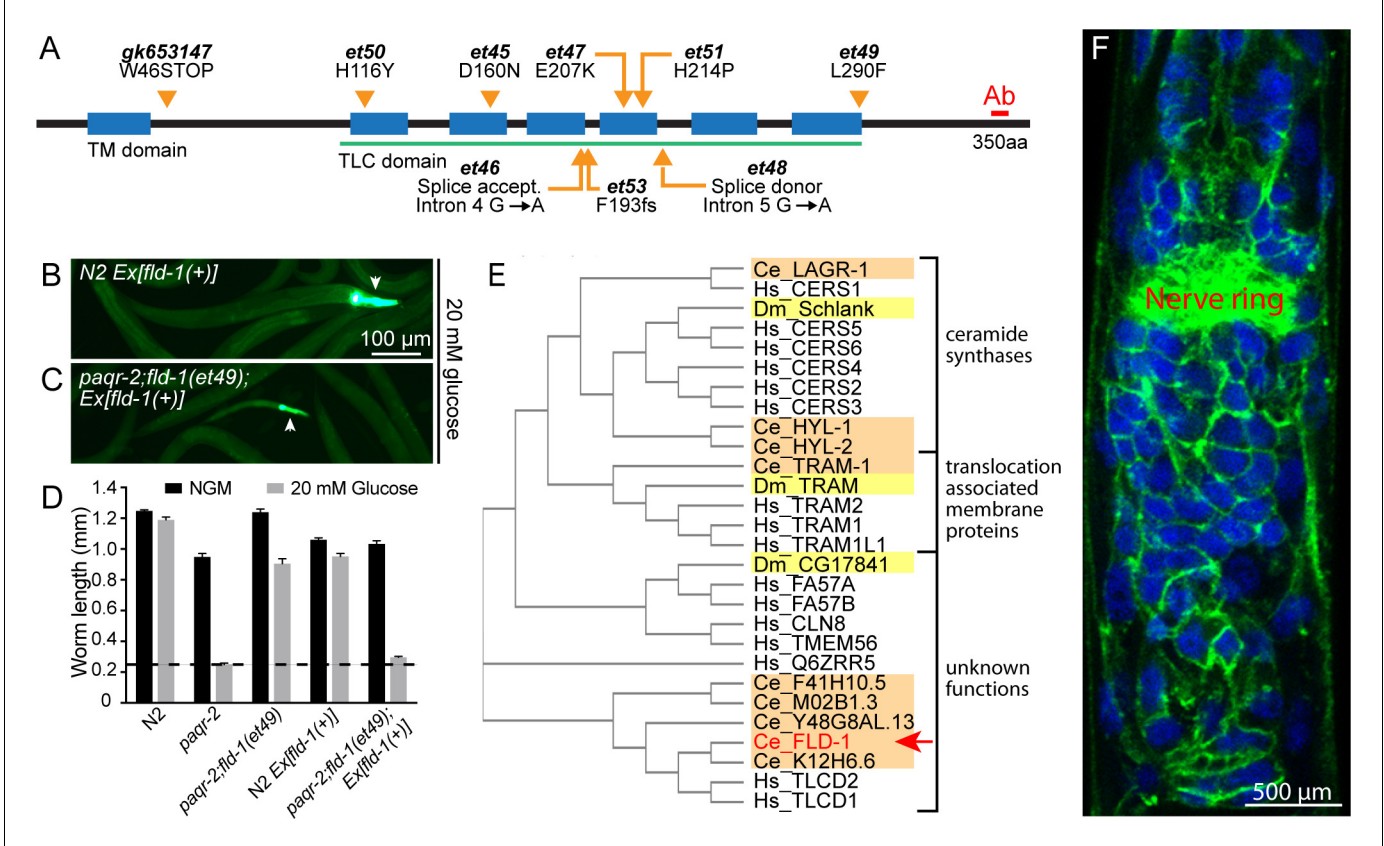

**Figure 1.** Mutant alleles, homology comparisons and expression profile of *C.elegans fld-1*. (**A**) Positions and effects of mutant *fld-1* alleles; transmembrane domains predicted by Phobius are indicated as blue boxes, and the Pfam PF03798 TLC domain is underlined in green. (**B–D**) Introduction of a wild-type *fld-1(+)* transgene in control N2 worms has no effect on their growth but restores glucose intolerance in the *paqr-2(tm3410); fld-1(et49)* double mutant, which shows that *fld-1(et49)* is a loss-of-function mutation that protects *paqr-2* mutant worms from glucose toxicity; transgenic worms are positive for the pharyngeal GFP marker. The dashed line indicates the approximate length of the L1s at the start of the experiment (~250 μm). Note that the experiments involving transgenes were performed separately. (**E**) Cladogram of the *C. elegans* (orange shading), *D. melanogaster* (yellow shading) and human (no shading) proteins homologous to the *C. elegans* FLD-1 protein (red text and arrow) as per Clustal W. (**F**) Confocal image of the anterior portion of a fixed *C. elegans* L1 larva expressing the in vivo translational reporter *Pfld-1::FLD-1::GFP*. This reporter is ubiquitously expressed in the plasma membrane (green) while the DAPI staining (blue) decorates most of the cytoplasm in each cell since fixation allowed the chromosomal DNA to escape the nucleus.

DOI: https://doi.org/10.7554/eLife.40686.003

The following figure supplements are available for figure 1:

**Figure supplement 1.** Description of the novel *fld-1* mutant alleles.
DOI: https://doi.org/10.7554/eLife.40686.004

**Figure supplement 2.** Alignment of Y63D3A.8 (i.e. FLD-1), its closest *C. elegans* homolog (K12H6.6) and two closest human homologs (TLCD1 and TLCD2).
DOI: https://doi.org/10.7554/eLife.40686.005

**Figure supplement 3.** Effect of independently isolated mutant alleles of *fld-1*, *Y48G8AL.13* and *F41H10.5*.
DOI: https://doi.org/10.7554/eLife.40686.006

**Figure supplement 4.** Expression profile of the in vivo reporter *Pfld-1::GFP*.
DOI: https://doi.org/10.7554/eLife.40686.007

**Figure supplement 5.** Tissue-specific expression of *FLD-1::GFP*.
DOI: https://doi.org/10.7554/eLife.40686.008

**Figure supplement 6.** *fld-1* single mutant worms have no obvious phenotype.
DOI: https://doi.org/10.7554/eLife.40686.009

protein family (at least 70 homologs) that share sequence homology in their multi-pass transmembrane domain: TRAM (translocating chain-associated membrane protein, important for the translocation of transmembrane proteins during their synthesis in the rough ER), Lag1p (an ER-resident protein that facilitates ER to Golgi transport of glycosylphosphatidylinositol-anchored proteins) and CLN8 (a ubiquitously expressed ER and Golgi-localized protein that causes neurological disorders when mutated) (*Winter and Ponting, 2002*). The function of TLCD1 and TLCD2 is not known, although TLCD1 (also known as calfacilitin) has been described as a positive regulator of calcium channel activity (*Papanayotou et al., 2013*). The same study localized TLCD1 to the plasma membrane of mammalian cells using a careful series of myc-epitope tagging and immunodetection experiments (*Papanayotou et al., 2013*). Transgenic *C. elegans* carrying a translational GFP reporter (*Pfld-1::FLD-1::GFP*) reveal that FLD-1 is also expressed in the plasma membrane of most or all cells throughout development and in adults (*Figure 1F* and *Figure 1—figure supplements 3C* and *4*). Note that the *Pfld-1::FLD-1::GFP* reporter is functional since it restores the glucose intolerance of the *paqr-2(tm3410)* mutant (*Figure 1—figure supplement 5A–B*). Expression of *FLD-1::GFP* in either hypodermis or intestine was also sufficient to restore the glucose intolerance of the *paqr-2(tm3410)* mutant, suggesting that *fld-1* can act in a variety of tissues (*Figure 1—figure supplement 5A and C*). This is consistent with an earlier study demonstrating that the maintenance of membrane homeostasis is cell nonautonomous and may rely on effective trafficking of lipids among tissues (*Bodhicharla et al., 2018*).

## *fld-1* mutations suppress several membrane-related defects

The *fld-1* single mutants have no obvious phenotypes and were indistinguishable from wild-type worms for the following traits: brood size, pharyngeal pumping, locomotion, defecation, survival at 37°C or 30°C, length, UPR$^{er}$ activation scored using the reporter *phsp-4::GFP* on normal media or in presence of glucose, and life span (*Figure 1—figure supplement 6A–I*). The only phenotype that we noted in the single *fld-1* mutant is an increased activation of the oxidative stress response reporter *gst-4::GFP* when the worms are cultivated in the presence of the LCPUFA linoleic acid (LA) (*Figure 1—figure supplement 6J–L*), suggesting a defect in LCPUFA management in the mutant. We also compared five of the novel *fld-1* alleles for their ability to suppress three previously described phenotypes caused by membrane homeostasis defects in the *paqr-2* mutant, namely glucose intolerance, cold intolerance and a withered tail tip morphology defect (*Devkota et al., 2017*; *Svensk et al., 2016a*; *Svensk et al., 2013*; *Svensson et al., 2011*; *Svensk et al., 2016b*). All five tested *fld-1* alleles could suppress the three phenotypes to a similar degree, indicating that they are all likely null alleles (*Figure 2A–B*). The *fld-1(et48)* allele, which carries a mutation in a splice donor site (*Figure 1A*), was chosen as the reference allele and used in most subsequent experiments. The *fld-1(et48)* mutation also suppresses the glucose intolerance of the *iglr-2(et34)* single mutant and *paqr-2(tm3410) iglr-2(et34)* double mutant (*Figure 2C*), as anticipated from the fact that *paqr-2* and *iglr-2* are mutually dependent for maintenance of membrane homeostasis (*Svensk et al., 2016a*). There exist four other genes in *C. elegans* that are closely related to *fld-1* (*Figure 1E*), and null mutants were available for two of them (*Y48G8AL.13* and *F41H10.5*); neither of these mutants suppressed the glucose intolerance of the *paqr-2(tm3410)* mutant (*Figure 1—figure supplement 3D*), but both provided some suppression of the cold intolerance defect (*Figure 1—figure supplement 3E*), suggesting that they may be partially redundant with *fld-1*.

## *fld-1* acts in a pathway distinct from other *paqr-2* suppressors

To try and elucidate the mechanism by which *fld-1(et48)* promotes membrane fluidity, we performed genetic interaction experiments with other *paqr-2* suppressors and also analysed the fatty acid composition in these mutants. The *fld-1(et48)* mutation greatly enhanced the effect of the *mdt-15(et14)*, *cept-1(et10)* and *hacd-1(et12)* mutations in supressing intolerance to glucose and, in the case of *mdt-15(et14)* and *cept-1(et10)*, to the more severe membrane-rigidifying diet of palmitic acid (PA)-loaded *E. coli* (*Figure 2D–F*). This result strongly indicates that *fld-1(et48)* acts in a pathway separate from that of *mdt-15* (a mediator subunit), *cept-1* (a choline/ethanolamine phosphotransferase) and *hacd-1* (a hydroxyl-acyl-CoA dehydrogenase), which promote membrane fluidity by activating FA desaturases (*Svensk et al., 2013*). In vivo FRAP measurements confirm that the *fld-1(et48)* allele restores membrane fluidity to *paqr-2(tm3410)* mutant worms cultivated in the presence of glucose

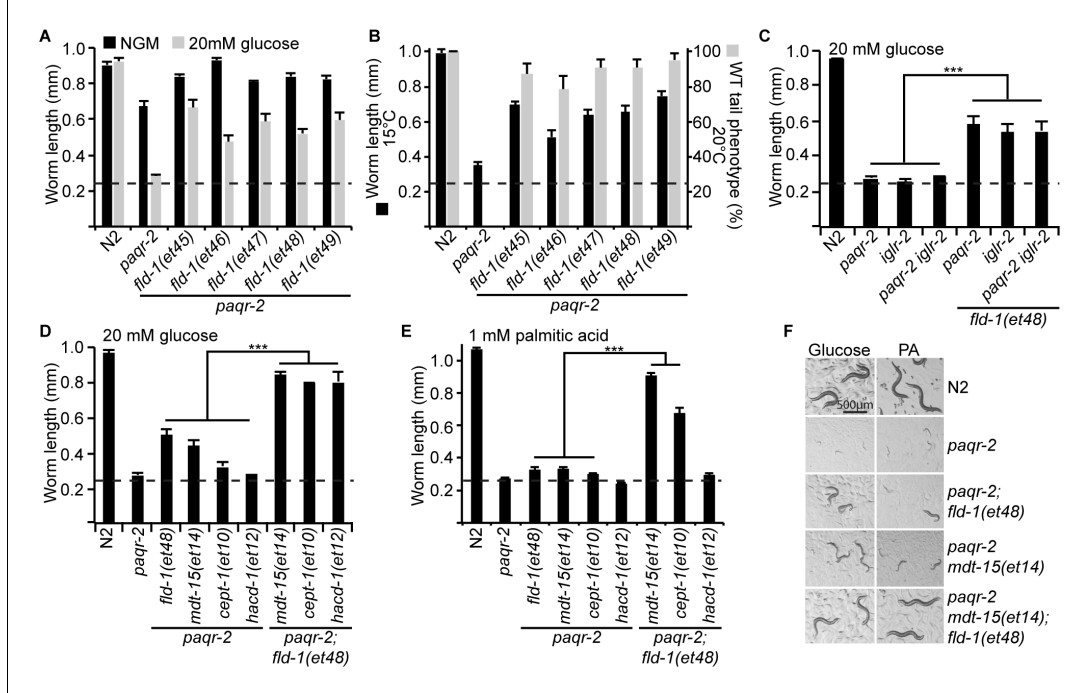

**Figure 2.** Comparison and genetic interaction of the novel *fld-1* alleles with other *paqr-2* and *iglr-2* suppressors. (A) The five *fld-1* alleles tested improve the growth of the *paqr-2* mutant on nematode growth media (NGM) and suppressed the *paqr-2* growth arrest phenotype on NGM media containing 20 mM glucose. (B) The five *fld-1* alleles tested suppressed the 15°C growth arrest and the 20°C tail tip defect phenotypes of the *paqr-2* mutant. (C) The reference *fld-1(et48)* allele suppresses the growth arrest phenotype of *paqr-2*, *iglr-2* and *paqr-2 iglr-2* single and double mutants grown on 20 mM glucose. (D) *fld-1(et48)* greatly enhances the suppression of the growth arrest phenotype of the *paqr-2* mutant cultivated on 20 mM glucose by the *mdt-15(et14)*, *cept-1(et10)* and *hacd-1(et12)* mutations, which suggest that it acts in a separate pathway. (E) *fld-1(et48)* greatly enhances the suppression of the growth arrest phenotype of the *paqr-2* mutant fed PA-loaded *E. coli* by the *mdt-15(et14)* and *cept-1(et10)* mutations which suggests that it acts in a separate pathway; *hacd-1(et12)* did not suppress the growth defect on PA-loaded *E. coli* even when combined with the *fld-1(et48)* mutation, suggesting that it is a weaker suppressor of *paqr-2* mutant phenotypes. (F) Representative images of worms from the experiment in panel (E). The dashed lines indicate the approximate length of the L1s at the start of the experiment (~250 μm).
DOI: https://doi.org/10.7554/eLife.40686.010

or PA-loaded *E. coli* (*Figure 3A–B*), and expression of the desaturase *fat-7::GFP* reporter was enhanced by *mdt-15(et14)* and *cept-1(et10)* but not by *fld-1(et48)* or *fld-1(et49)*, again showing that *fld-1* acts in a separate pathway (*Figure 3C–D*). Interestingly, RNAi knockdown of the desaturases *fat-6* and/or *fat-7* abolished the suppression of *paqr-2(tm3410)* glucose intolerance by *fld-1(et48)*, and this was the case both in the presence or absence of the *mdt-15(et14) gof* allele (*Figure 3E*). Knockdown of the desaturase *fat-5* also abolished the suppression of *paqr-2(tm3410)* glucose intolerance by *fld-1(et48)* but not in the presence of *mdt-15(et14)* which likely drives sufficient compensatory induction of *fat-6* and *fat-7*. Taken together, these results indicate that *fld-1(et48)* does not act by promoting increased desaturase expression but that desaturase activity is important for its effect on membrane homeostasis. Phosphatidylethanolamines (PEs) are the most abundant phospholipids in *C. elegans* (*Dancy et al., 2015*). An analysis of the PE composition in worms challenged with growth on glucose shows that the *paqr-2(tm3410)* mutant has an excess of SFAs and reduced levels of LCPUFAs, and that these defects are partially corrected by the *mdt-15(et14)* or *fld-1(et48)* mutations, and fully corrected by a combination of both these mutations (*Figure 3F–G*). The levels of monounsaturated fatty acids (MUFAs) were unchanged in the PEs of the *fld-1* mutant, even in combination with the *mdt-15(et14)* mutation (*Figure 3—figure supplement 1A*). Similar results were obtained when using PA-loaded *E. coli* as a rigidifying challenge rather than glucose (*Figure 3—figure supplement 1B–C*). Note that the lipidomic analysis was performed on whole worms, and thus reflects global lipid composition rather than the composition of any specific membrane; also, cholesterol levels were not measured since this lipid does not appear to play a structural role in *C. elegans* but instead is a likely precursor for essential low-abundance metabolites (*Merris et al., 2004*).

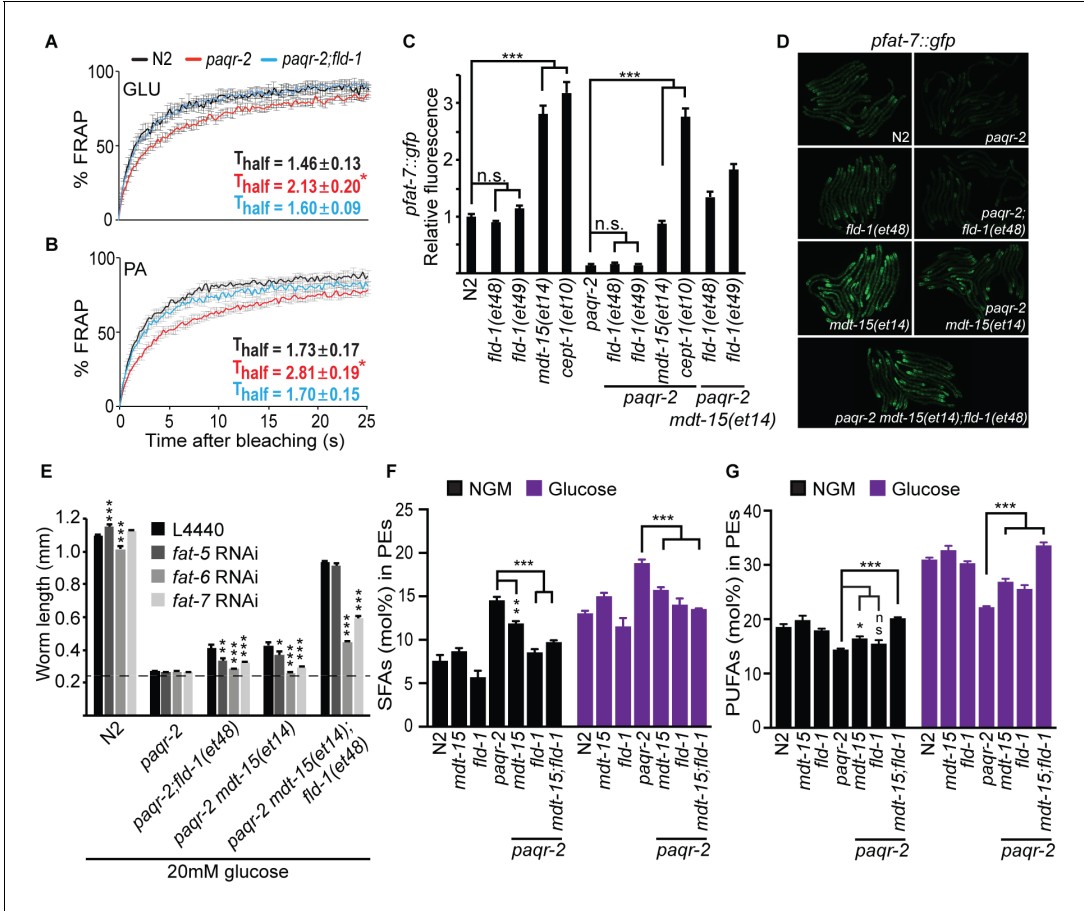

**Figure 3.** *fld-1(et48)* corrects membrane phospholipid composition and several membrane defects of the *paqr-2* mutant. (**A–B**) In vivo FRAP measurements show that *fld-1(et48)* suppresses the membrane fluidity defect in *paqr-2* mutant worms grown on 20 mM glucose or fed PA-loaded *E. coli*, respectively; note the normalization of the $T_{half}$ in *paqr-2;fld-1* double mutants. (**C–D**) The *fld-1(et48)* and *fld-1(et49)* mutations do not induce upregulation of the *pfat-7::gfp* reporter in wild-type or *paqr-2* mutant worms; in contrast, the *mdt-15(et14)* and *cept-1(et10)* mutations do cause *pfat-7::gfp* upregulation in both wild-type and *paqr-2* mutant worms. The combination of *mdt-15(et14)* with *fld-1(et48)* or *fld-1(et49)* caused a slight increase in *pfat-7::gfp* in *paqr-2* mutants compared to the effect of *mdt-15(et14)* alone, likely reflecting their improved health (see **Figure 2D**). (**E**) RNAi against the Δ9 desaturases *fat-6* and *fat-7* impairs the suppression of the *paqr-2* mutant growth arrest on 20 mM glucose by *fld-1(et48)*, *mdt-15(et14)* or *mdt-15 (et14);fld-1(et48)*; RNAi against *fat-5* has a weaker effect. The dashed line indicates the approximate length of the L1s at the start of the experiment (~250 µm). (**F–G**) The *paqr-2* mutant shows an excess of SFAs and reduced LCPUFAs levels among PEs on control plates and 20 mM glucose plates, and this is partially normalized by the *mdt-15(et14)* and *fld-1(et48)* single mutations and fully or nearly fully normalized by combining both mutations.

DOI: https://doi.org/10.7554/eLife.40686.011

The following source data and figure supplements are available for figure 3:

**Source data 1.** Lipidomics data for panel F.
DOI: https://doi.org/10.7554/eLife.40686.020
**Source data 2.** Lipidomics data for panel G.
DOI: https://doi.org/10.7554/eLife.40686.021
**Figure supplement 1.** The *fld-1(et48)* mutation ameliorates the *paqr-2* mutant lipid profiles and enhances the protective effects of *mdt-15(et14)*.
DOI: https://doi.org/10.7554/eLife.40686.012
**Figure supplement 1—source data 1.** Lipidomics data for panel A.
DOI: https://doi.org/10.7554/eLife.40686.013
**Figure supplement 1—source data 2.** Lipidomics data for panel B.
DOI: https://doi.org/10.7554/eLife.40686.014
**Figure supplement 1—source data 3.** Lipidomics data for panel C.
DOI: https://doi.org/10.7554/eLife.40686.015
**Figure supplement 1—source data 4.** Lipidomics data for panel D.
DOI: https://doi.org/10.7554/eLife.40686.016

*Figure 3 continued on next page*

*Figure 3 continued*

**Figure supplement 2.** The *fld-1(et48)* mutation protects *paqr-2* mutants against membrane rigidification.
DOI: https://doi.org/10.7554/eLife.40686.017

**Figure supplement 3.** The *fld-1(et48)* mutation suppresses the growth defect of the *fat-2(wa17)* mutant and promotes accumulation of the LCPUFA eicosapentaenoic acid (EPA).
DOI: https://doi.org/10.7554/eLife.40686.018

**Figure supplement 3—source data 1.** Lipidomics data for panel D.
DOI: https://doi.org/10.7554/eLife.40686.019

Altogether, these results are consistent with the *fld-1(et48)* mutation acting in a pathway independent of *mdt-15* to promote LCPUFA levels in phospholipids, at the expense of SFAs, hence increasing membrane fluidity. Some mechanism likely exists to prevent excess membrane fluidity since the *fld-1(et48)* and *mdt-15(et14)* mutations do not cause an excess fluidity either by themselves or in combination in any of the tested conditions (*Figure 3—figure supplement 2*).

## *fld-1* influences the levels of PUFA-containing phospholipids

Mechanistically, it is possible that the normal function of *fld-1* is either to limit the generation of LCPUFA-containing phospholipids or to promote their turnover. In either case, mutations in *fld-1* would not only suppress the fluidity defect of the *paqr-2* and *iglr-2* mutants but may additionally suppress the phenotypes of mutants where the production of PUFAs is inefficient. We tested this using the *fat-2* mutant, which is unable to convert 18:1 to 18:2, that is the primary precursor for LCPUFAs in *C. elegans* (*Watts and Ristow, 2017*). We found that, indeed, the *fld-1* mutation suppresses the growth defect of the *fat-2* mutant both on normal plates and plates containing 20 mM glucose, even in a *paqr-2(tm3410)* mutant background (*Figure 3—figure supplement 3A–C*). We interpret these results to mean that the *fld-1* mutation causes either increased production of LCPUFA-containing phospholipids or lowers their turnover; either effect therefore allows the abundance of LCPUFA-containing phospholipids to reach an increased level in membranes even if their production is reduced by mutations in *paqr-2*, *iglr-2* or *fat-2*. This interpretation is further supported by the observation that *fld-1(et48)* can suppress the inability of the *paqr-2* mutant to accumulate the exogenously provided LCPUFAs eicosapentaenoic acid (EPA; 20:5) (*Figure 3—figure supplement 3D*); notably, this demonstrates that no synthesis of LCPUFAs is required for *fld-1(et48)* to promote their accumulation among phospholipids.

## Mammalian TLCD1 and TLCD2 are regulators of membrane composition and fluidity

The mouse and human TLCD1 and TLCD2 genes are expressed in a variety of tissues, with heart, muscle, liver and fat being strongest for TLCD1 and heart, muscle, liver, small intestine and fat being strongest for the more variably expressed TLCD2 (*Fagerberg et al., 2014*) (*Figure 4—figure supplement 1*). The molecular function of these proteins in mammals appears conserved with that of the *C. elegans fld-1*: knockdown of either gene using siRNA in human HEK293 cells has little effect on these cells under normal conditions (*Figure 4—figure supplement 2A–D*) but was protective against the membrane rigidifying effects of PA (*Figure 4A–B* and *Figure 4—figure supplement 2E*); knockdown of the control gene PPIB had no effect on membrane fluidity (*Figure 4—figure supplement 2F–G*). Importantly, knockdown of TLCD1 and TLCD2 also protected the HEK293 cells from PA-induced apoptosis (*Figure 4—figure supplement 2H–I*).

Several mechanisms could explain the protective effects of TLCD1/2 knockdown, many of which could be experimentally ruled out. TLCD1 and TLCD2 do not regulate the uptake of exogenous SFAs since there is no difference in the uptake rate of labelled PA in HEK293 cells treated with siRNA against either gene (*Figure 4—figure supplement 2J*). The SFA content in the TAGs of PA-treated HEK293 cells is also unaffected by TLCD1 or TLCD2 knockdown (*Figure 4—figure supplement 2K*), as is their TAG/PC ratio (*Figure 4—figure supplement 2L*) and Nile red staining of lipid stores (*Figure 4—figure supplement 2M–N*), suggesting no effect on either uptake or sequestering of FAs. The expression levels of the desaturases SCD and FADS1-3 are also not increased by TLCD1 or TLCD2 knockdown (*Figure 4—figure supplement 2O*) ruling this out as a mechanism for the

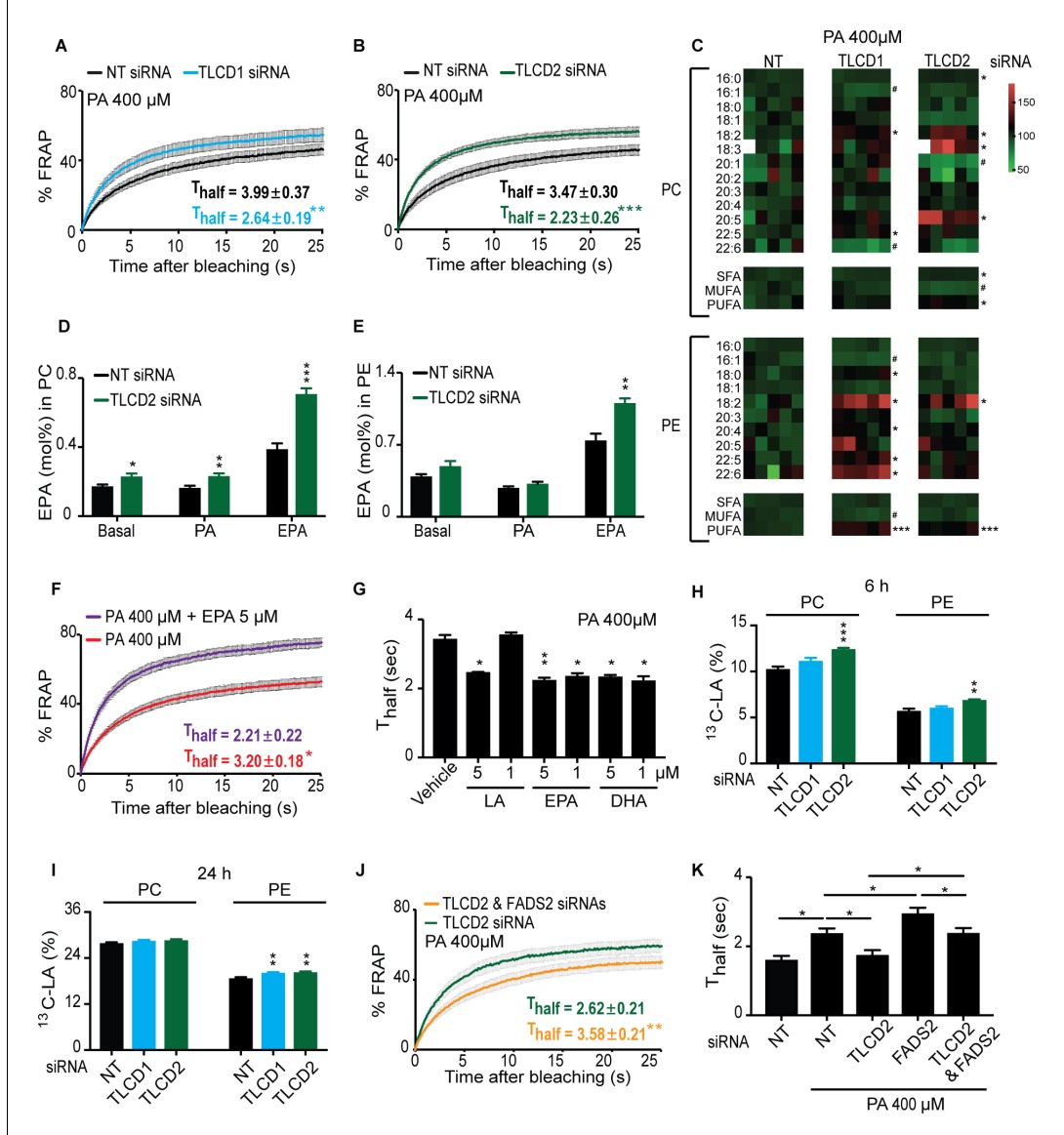

**Figure 4.** The mammalian TLCD1 and TLCD2 proteins regulate membrane composition and fluidity. (A–B) FRAP analysis showing that siRNA against TLCD1 or TLCD2 prevent membrane rigidification by 400 μM PA in HEK293 human cells; note the significantly lower $T_{half}$ value in the TLCD1 and TLCD2 siRNA samples. (C) Heat map showing the relative abundance of various FAs in PCs and PEs of HEK293 cells treated with siRNA against TLCD1 or TLCD2 and challenged with 400 μM PA; the data are normalized to the non-target siRNA (NT) average value for each FA. Note the strong increase in several individual LCPUFAs (e.g. 18:2 and 22:6) among PEs in TLCD1 siRNA-treated cells and in LCPUFAs among both PCs (e.g. 18:2, 18:3 and 20:5) and PEs (e.g. 18:2) among TLCD2 siRNA-treated cells. TLCD2 siRNA causes an increased accumulation of EPA in PCs under basal, 400 μM PA-treated or EPA-treated conditions (D), and also in PEs when the HEK293 cells are treated with 0.9 μM EPA (E). (F–G) 1 μM EPA or DHA prevents membrane rigidification by 400 μM PA in HEK293 cells, while 5 μM linoleic acid (LA) is required for the same effect. (H–I) Amount of LA-[$^{13}C_{18}$] present in PCs and PEs after 6 and 24 hr of incubation, respectively, with cells treated with the indicated siRNA (NT, TLCD1 or TLCD2). (J–K) Knockdown of the desaturase FADS2 reduces the membrane fluidity of HEK293 cells cultivated in 400 μM PA but does not prevent the protective effect of TLCD2 knockdown.

DOI: https://doi.org/10.7554/eLife.40686.022

The following video, source data, and figure supplements are available for figure 4:

**Source data 1.** Lipidomics data for panel C.
DOI: https://doi.org/10.7554/eLife.40686.031
**Source data 2.** Lipidomics data used in panels D-E.
DOI: https://doi.org/10.7554/eLife.40686.032
**Source data 3.** Lipidomics data for panels H-I.
DOI: https://doi.org/10.7554/eLife.40686.033

*Figure 4 continued on next page*

*Figure 4 continued*

**Figure supplement 1.** Relative expression levels of TLCD1 and TLCD2 among various mouse tissues.
DOI: https://doi.org/10.7554/eLife.40686.023
**Figure supplement 2.** Specificity of the TLCD1/2 siRNA effects.
DOI: https://doi.org/10.7554/eLife.40686.024
**Figure supplement 2—source data 1.** Lipidomics data for panels K-L.
DOI: https://doi.org/10.7554/eLife.40686.025
**Figure supplement 3.** Effect of TLCD1/2 knockdown on FA composition and membrane fluidity.
DOI: https://doi.org/10.7554/eLife.40686.026
**Figure supplement 3—source data 1.** Lipidomics data for panels A-B.
DOI: https://doi.org/10.7554/eLife.40686.027
**Figure supplement 3—source data 2.** Lipidomics data for panels C-D.
DOI: https://doi.org/10.7554/eLife.40686.028
**Figure supplement 3—source data 3.** Lipidomics data for panels E-H.
DOI: https://doi.org/10.7554/eLife.40686.029
**Figure supplement 3—source data 4.** Lipidomics data for panels I-J.
DOI: https://doi.org/10.7554/eLife.40686.030
**Figure 4–video 1.** Movie of a FRAP experiment on NT siRNA-treated HEK293 cells.
DOI: https://doi.org/10.7554/eLife.40686.034
**Figure 4–video 2.** Movie of a FRAP experiment on NT siRNA-treated HEK293 cells cultivated in the presence of 400 µM PA.
DOI: https://doi.org/10.7554/eLife.40686.035
**Figure 4–video 3.** Movie of a FRAP experiment on TLCD2 siRNA-treated HEK293 cells cultivated in the presence of 400 µM PA.
DOI: https://doi.org/10.7554/eLife.40686.036

fluidizing effect; indeed, the SCD expression is downregulated in TLCD1 or TLCD2 knockdown which may reflect a decreased demand on their activity.

The most likely mechanisms of action for TLCD1/2 is that they either act by limiting the production of LCPUFA-containing phospholipids or by promoting their turnover, thus echoing our speculations regarding FLD-1 function in *C. elegans.* This conclusion is supported by the marked increase in the abundance of the 18:2, 20:5 and 22:6 LCPUFAs in the PEs of TLCD1 knockdown cells, and of the 18:2, 18:3 and 20:5 LCPUFAs in the PCs of TLCD2 knockdown cells (*Figure 4C*). No other major changes were observed in the lipid profiles of TLCD1/2 knockdown cells. Specifically, there were no large differences in the FA composition of PCs or PEs under normal conditions (*Figure 4—figure supplement 3A–B*), in the PC/PE ratio and cholesterol content under basal or PA-challenged conditions (*Figure 4—figure supplement 3C–D*), or in the ceramide levels which were dramatically increased upon cultivation in PA (*Figure 4—figure supplement 3E–H*). Note that, as in *C. elegans,* the lipidomic analysis was performed on whole cells, and thus reflects global lipid composition rather than the composition of any specific membrane.

Cultivating HEK293 cells in the presence of exogenous EPA leads to its excess accumulation in PCs and PEs when TLCD2 is knocked down (*Figure 4D–E*), which demonstrates that, as in *C. elegans*, de novo PUFA synthesis is not required for the TLCD2 effect on PUFA-containing phospholipids; no change in TAG storage or EPA storage in TAGs occurred upon TLCD2 knockdown, indicating that TLCD2 does not regulate fatty acid storage or LCPUFA uptake (*Figure 4—figure supplement 3I–J*). The effect of TLCD2 knockdown on phospholipid EPA levels is especially interesting since LCPUFAs are very potent fluidizing lipids (*Yang et al., 2011*), which we confirmed experimentally in our cell system where as little as 1 µM of EPA or docosahexaenoic acid (DHA) prevents membrane rigidification by 400 µM PA (*Figure 4F–G*). Note that caution should be taken when quantitatively interpreting these types of experiments since the uptake, metabolism, incorporation, subcellular localization and turnover of the lipids being compared may differ substantially.

We then considered whether the TLCDs may act by limiting the generation of LCPUFA-containing phospholipids or by promoting their turnover. To distinguish between these two hypotheses, we incubated HEK293 cells in the presence of the [13]C-labeled LCPUFA LA (18:2) and monitored its incorporation and clearance rate in control cells or cells where either TLCD1 or TLCD2 had been knocked down. The results show that TLCD2 knockdown caused increased incorporation of LA in PCs and PEs within 6 hr of incubation, and that increased incorporation was also evident by 24 hr in

the PEs of TLCD1 and TLCD2 knockdown cells (*Figure 4H–I*). No obvious differences were observed in the TLCD1/2 knockdown cells during the chase part of the experiment (*Figure 4—figure supplement 3K–L*). Altogether, these results indicate that the primary function of TLCD1 and TLCD2 is to limit the formation of LCPUFA-containing phospholipids rather than promote their turnover.

Finally, we further confirmed functional conservation between *C. elegans* FLD-1 and mammalian TLCD1/2 in several ways. Firstly, we found that TLCD2 siRNA protects against the potentiated rigidifying effects of PA when the FADS2 desaturase is knocked down (*Figure 4J–K*; *Figure 4—figure supplement 3M*), just as the *fld-1* mutation suppressed the effects of the *fat-2* mutation in *C. elegans*; this also strengthens the conclusion that TLCD2 acts independently from the desaturases. We also found that TLCD2 knockdown suppresses membrane rigidification in HEK293 cells where AdipoR2 had been knocked down, which by itself increases the sensitivity to the effect of PA on membrane rigidification (*Devkota et al., 2017*) (*Figure 5A–B*; *Figure 5—figure supplement 1A*). This again echoes the *C. elegans* findings where *fld-1 lof* mutations suppress the membrane rigidity phenotype of the *paqr-2* mutant. As noted earlier, TLCD2 knockdown causes a strong increase in the levels of EPA in PCs (*Figure 4C*), which is a potent fluidizing LCPUFA: 5 µM are sufficient to

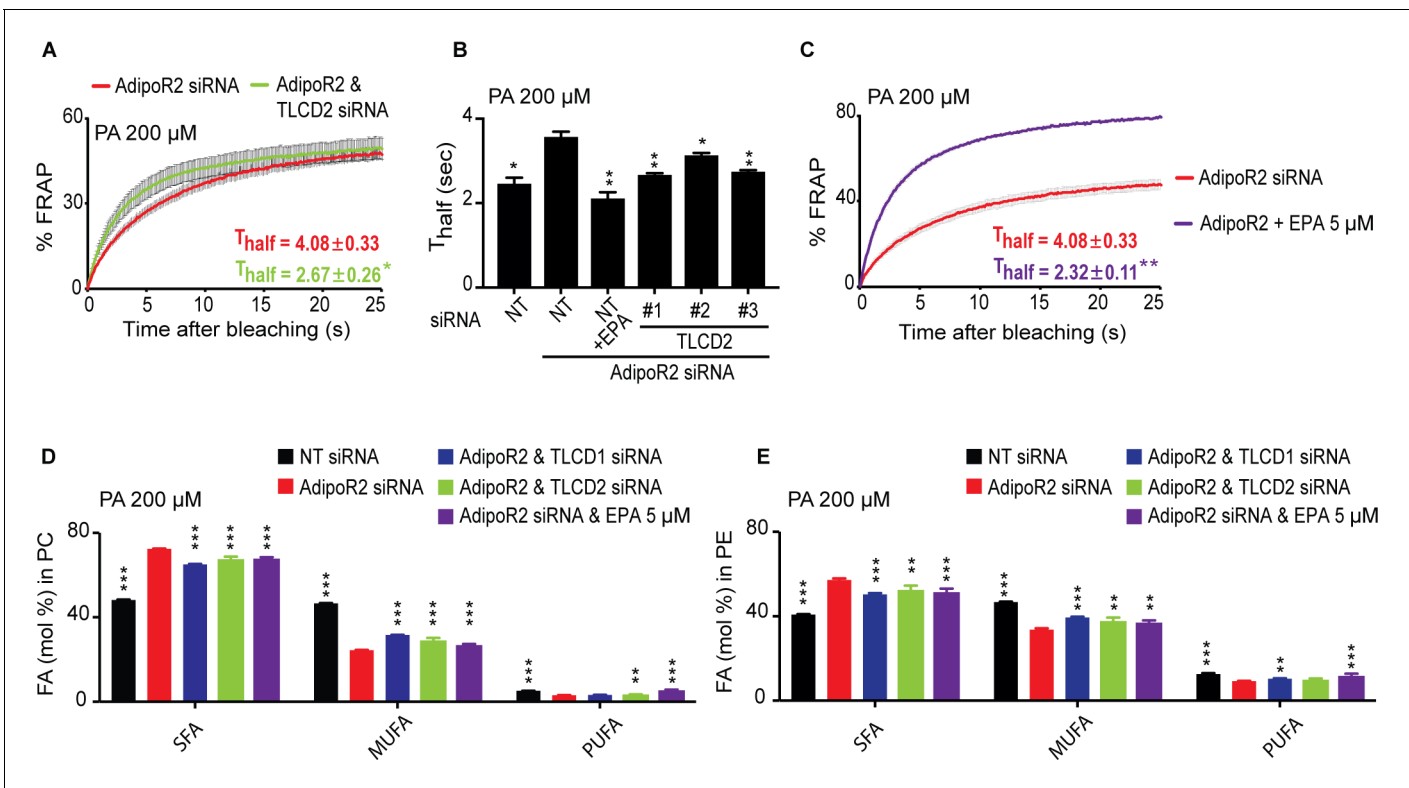

**Figure 5.** TLCD2 knockdown protects against AdipoR2 knockdown. (**A–C**) FRAP analysis showing that siRNA against TLCD2 or inclusion of 5 µM EPA protects against the rigidifying effects of 200 µM PA in cells where AdipoR2 is also knocked down. Note the significantly lower $T_{half}$ value in the TLCD2 siRNA and EPA-treated samples. (**D–E**) AdipoR2 siRNA causes an excess of SFAs and depletion of MUFAs and PUFAs in the PCs and PEs of HEK293 cells; these effects of AdipoR2 knockdown are partially abrogated by siRNA against TLCD2 or by the inclusion of 5 µM EPA.
DOI: https://doi.org/10.7554/eLife.40686.037

The following source data and figure supplements are available for figure 5:

**Source data 1.** Lipidomics data used in panel D.
DOI: https://doi.org/10.7554/eLife.40686.040

**Source data 2.** Lipidomics data for panel E.
DOI: https://doi.org/10.7554/eLife.40686.041

**Figure supplement 1.** Protective effects of TLCD1/2 knockdown against AdipoR2 knockdown.
DOI: https://doi.org/10.7554/eLife.40686.038

**Figure supplement 1—source data 1.** Lipidomics data used in panels B-E
DOI: https://doi.org/10.7554/eLife.40686.039

completely protect AdipoR2 knockdown cells from the rigidifying effects of 200 µM PA (*Figure 5C*). TLCD2 knockdown also suppresses the composition defect in HEK293 cells where AdipoR2 had been knocked down: TLCD1 and TLCD2 siRNA both improved the UFA content of PCs and PEs in AdipoR2 knockdown cells treated with 200 µM PA (*Figure 5D–E*), and also normalized ceramide levels in these cells (*Figure 5—figure supplement 1B–E*).

## Discussion

The fatty acid composition of cellular membranes reflects the composition of the dietary fats. This is especially evident for complex dietary PUFAs that become incorporated into membrane phospholipids in *C. elegans* (*Dancy et al., 2015*) and in mammals (*Pan and Storlien, 1993*; *Abbott et al., 2012*; *Andersson et al., 2002*). However, regulatory mechanisms must exist to adjust membrane composition, hence properties, in response to diets with a wide range of SFA/UFA ratios. The present study shows that *fld-1* in *C. elegans*, and TLCD1/2 in mammalian cells, potentiate the toxicity of exogenous SFAs, and act by limiting the generation of LCPUFA-containing phospholipids (see the model in *Figure 6*). The net result of mutations in *fld-1* or knockdown of TLCD1/2 is therefore an increase in LCPUFA-containing phospholipids, which helps maintain membrane fluidity in the presence of exogenous SFAs. In particular, LCPUFAs, which are present in elevated levels in the worm *fld-1* mutant and in human HEK293 cells with knockdown of either TLCD1 or TLCD2, are potent membrane fluidizers, as we have shown here (*Figure 4F–G* and *Figure 5C*) and in agreement with other studies (*Yang et al., 2011*). Hence, a small increase in LCPUFAs can greatly increase membrane fluidity.

Structurally, the FLD-1 and TLCD1/2 proteins are characterized by the presence of a TLC domain and are therefore distantly related to the ER-localized ceramide synthases (human CERS1-6) and to the translocation associated membrane proteins (human TRAM1, TRAM2 and TRAM1L1) involved in membrane protein synthesis in the rough ER (*Pewzner-Jung et al., 2006*; *Winter and Ponting, 2002*). Other TLC domain-containing proteins exist (see *Figure 1B*) but their function remains

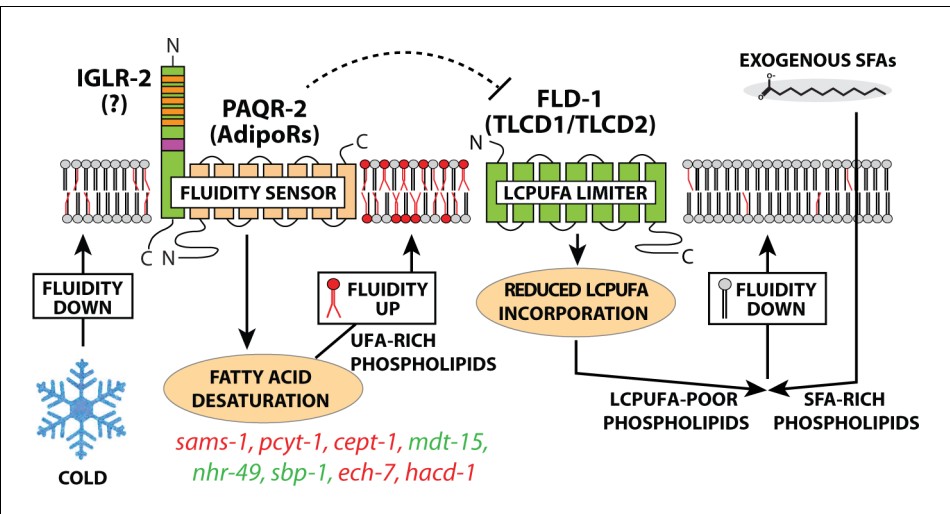

**Figure 6.** Updated model of membrane fluidity regulation in *C. elegans*. Loss of membrane fluidity, which may result from lowered temperature or SFA-rich diets, is sensed by the PAQR-2/IGLR-2 complex that then signals to promote FA desaturation, hence restoring fluidity; *lof* and *gof* mutations in the genes labelled in red and green, respectively, can act as *paqr-2/iglr-2* suppressors (*Devkota et al., 2017*; *Svensk et al., 2016a*; *Svensk et al., 2013*). The present study shows that an alternative route to promote membrane fluidity is to inhibit FLD-1, for example by mutation, which results in reduced turnover of LCPUFA-containing phospholipids, hence allowing their accumulation. The names of mammalian homologs for PAQR-2 and FLD-1 are indicated in parenthesis; the mammalian homolog of IGLR-2 is not yet identified. The possibility that PAQR-2 regulates FLD-1 (dashed line) is suggested by the fact that *lof* mutations in *fld-1* suppress *paqr-2* mutant phenotypes; there is at present no evidence for physical interactions.
DOI: https://doi.org/10.7554/eLife.40686.042

unclear, though some are suggested to act in lipid sensing, transport or synthesis (*Winter and Ponting, 2002*). In contrast to the CERSs and TRAMs, FLD-1 and TLCD1/2 are localized to the plasma membrane rather than in the ER (*Figure 1F* and *Figure 1—figure supplement 4*) (*Papanayotou et al., 2013*). This, and the fact that they lack critical motifs, indicate that FLD-1 and TLCD1/2 are neither ceramide synthases nor translocation associated proteins. Instead, their localization and effect on PUFA content in membrane phospholipids suggest that they act at the level of the plasma membrane by limiting the amounts of LCPUFA-containing phospholipids. It is possible, for example, that FLD-1 and TLCD1/2 influence substrate selection by phospholipases or lysophospholipid acyltransferases, that is regulating the Lands cycle through which phospholipids are actively remodelled by fatty acid exchange and hence effectively influence membrane composition and properties (*Shindou et al., 2017*). In particular, the presence of several membrane-bound acyl transferases in *C. elegans* suggest a promising avenue of genetic interaction studies to test their possible connection to FLD-1 activity (*Matsuda et al., 2008*; *Lee et al., 2008*). Note however that the plasma membrane localisation of FLD-1 and TLCD1/2 makes a direct control of lysophospholipid acyltransferases, which are believed to localize in internal membranes, notably the ER, rather unlikely.

The type of LCPUFAs that are elevated in the *C. elegans fld-1* mutant and in HEK293 treated with siRNA against TLCD1/2 could have other effects besides influencing membrane fluidity. In particular LCPUFAs can serve as precursors for many signalling lipids (*Funk, 2001*; *Watts, 2016*). That *fld-1* has its primary effects on membrane fluidity is however supported by several lines of evidence, much of which hinges on the fact that *fld-1* mutants act as *paqr-2* and *iglr-2* suppressors. This is important because the *paqr-2* and *iglr-2* mutants clearly have membrane fluidity defects since: a) They are sensitive to membrane-rigidifying conditions (cold or exogenous SFAs); b) Their mutant phenotypes are suppressed by other mutations that promote an increase in fluidizing UFAs, by exogenous UFAs or by fluidizing amounts of mild detergents (NP-40 or Triton X-100); c) The *paqr-2* and *iglr-2* mutants have an excess of membrane-rigidifying SFA-containing phospholipids; and d) The *paqr-2* and *iglr-2* phenotypes are closely associated with reduced membrane fluidity measured in vivo using FRAP under numerous different experimental conditions or genotypes (*Devkota et al., 2017*; *Svensk et al., 2016a*; *Svensk et al., 2013*; *Svensson et al., 2011*). Similarly, siRNA against AdipoR2 causes an increase in phospholipid SFAs and reduced membrane fluidity in the presence of exogenous palmitate as determined using FRAP or the Laurdan dye method, and siRNA against TLCD1/2 suppresses this membrane fluidity defect.

The effect of TLCD1/2 on plasma membrane composition likely explains the published observations that TLCD1 facilitates the activity of calcium channels that are essential during neural plate development (*Papanayotou et al., 2013*). Indeed, membrane composition can have strong effects on a variety of processes including insulin secretion by beta cells (*MacDonald et al., 2015*), glucose transport (*Ginsberg et al., 1982*), endocytosis (*Krischer et al., 1993*; *Illinger et al., 1991*), regulation of metabolic rate (*Hulbert and Else, 2000*), platelet aggregation (*Winocour et al., 1990*) and TRPV channel activity (*Caires et al., 2017*). Conversely, given the chemical instability of PUFAs, which are readily peroxidized (*Gaschler and Stockwell, 2017*; *Ayala et al., 2014*; *Kelley et al., 2014*; *Catalá, 2013*), it seems natural for cells to have mechanisms that limit their levels. Indeed, we found that the *C. elegans fld-1* mutant exhibits an enhanced oxidative stress response when challenged with the LCPUFA LA (*Figure 1—figure supplement 6J–L*), which indicates a function for *fld-1* in preventing such PUFA-derived oxidative stress. The ubiquitous plasma membrane expression of *C. elegans fld-1* suggests that it functions in most cells, though it is not required under non-challenging conditions given the relative health of the mutant.

The present work further establishes that a function of *paqr-2* and *iglr-2* in *C. elegans* is to prevent SFA lipotoxicity, and specifically to prevent membrane rigidification by SFAs, and further suggests that they may act via two separate pathways: 1) Upregulation of desaturases to promote membrane fluidity; and 2) Inhibition of *fld-1* directly or of a *fld-1*-regulated pathway. This conclusion is supported by the existence of two classes of *paqr-2* suppressors: mutations that activate desaturases (e.g. *gof* mutations in *nhr-49* or *mdt-15*, and *lof* mutations in *sams-1*, *cept-1*, *pcyt-1*, *ech-7* or *hacd-1*) (*Svensk et al., 2013*) and loss-of-function mutations in *fld-1*, which does not act by upregulating desaturases though it depends on their activity (present work). The suggestion that *paqr-2* may influence *fld-1* activity is raised only as a possibility consistent with the observed genetic interactions; we have at present no biochemical evidence that these proteins interact with each other nor do we know of a mechanism by which *paqr-2* could regulate *fld-1*.

We propose that inhibitors of TLCD1/2 may have therapeutic potential in instances of lipotoxicity and excess membrane rigidity, which is a feature of the diabetic condition (*Pilon, 2016*; *Perona, 2017*), or in instances of mutations in AdipoR1 that cause retina defects (*Zhang et al., 2016*; *Rice et al., 2015*). Encouragingly, we found that the *fld-1 C. elegans* single mutant has no obvious phenotype, which suggests that inhibition of the mammalian homologs may be well tolerated.

# Materials and methods

**Key resources table**

| Reagent type (species) or resource | Designation | Source or reference | Identifiers | Additional information |
|---|---|---|---|---|
| Strain, strain background (*C. elegans*) | N2 | *C. elegans Genetics Center (CGC)* | | |
| Strain, strain background (*C. elegans*) | CL2166 (dvIs19 [(pAF15)gst-369 4 p::GFP::NLS] | C. elegans Genetics Center (CGC); PMID: 12078522 | | |
| Strain, strain background (*C. elegans*) | HA1842 [rtIs30(pfat-7::GFP)] | PMID: 22035958 | | Gift from Amy Walker |
| Genetic reagent (*C. elegans*) | fld-1(et45); fld-1(et46); fld-1(et47); fld-1(et48); fld-1(et49); fld-1(et50); fld-1(et51); fld-1(et53) | This paper | | *et45, et48 and et49* will be deposited at CGC |
| Genetic reagent (*C. elegans*) | SJ4005 [zcIs4 (hsp-4::GFP)] | *C. elegans Genetics Center (CGC)* | | |
| Genetic reagent (*C. elegans*) | fld-1(gk653147) | *C. elegans Genetics Center (CGC)* | | From strain VC40470 |
| Genetic reagent (*C. elegans*) | paqr-2(tm3410) | *C. elegans Genetics Center (CGC); doi:10.1371/ journal.pone.0021343* | | |
| Genetic reagent (*C. elegans*) | iglr-2(et34) | *C. elegans Genetics Center (CGC); PMID 27082444* | | |
| Genetic reagent (*C. elegans*) | mdt-15(et14) | *C. elegans Genetics Center (CGC); DOI 10.1371/journal. pgen.1003801.s013* | | |
| Genetic reagent (*C. elegans*) | cept-1(et10) | *C. elegans Genetics Center (CGC); DOI 10.1371/journal. pgen.1003801.s013* | | |
| Genetic reagent (*C. elegans*) | hacd-1(et12) | *C. elegans Genetics Center (CGC); DOI 10.1371/journal. pgen.1003801.s013* | | |
| Genetic reagent (*C. elegans*) | Y48G8AL.13 (ok3097) | *C. elegans Genetics Center (CGC)* | | |
| Genetic reagent (*C. elegans*) | F41H10.5 (gk530235) | *C. elegans Genetics Center (CGC)* | | From strain VC40240 |
| Cell line (*Homo sapiens*) | HEK293 | | | |

*Continued on next page*

*Continued*

| Reagent type (species) or resource | Designation | Source or reference | Identifiers | Additional information |
|---|---|---|---|---|
| Antibody | Rabbitt anti-FLD-1 | This paper | | Raised by GeneScript against this peptide: TQVGDVESGPLRTQ. Used at 1:500 dilution for Western blot. |
| Recombinant DNA reagent | *Pfld-1::FLD-1* | This paper | | |
| Recombinant DNA reagent | *Pfld-1::FLD-1::GFP* | This paper | | |
| Recombinant DNA reagent | *Pelt-3::FLD-1::GFP* | This paper | | |
| Recombinant DNA reagent | *Pges-1::FLD-1::GFP* | This paper | | |
| Sequence-based reagent | TLCD1 siRNA | Dharmacon | J-015483–10 | |
| Sequence-based reagent | TLCD2 siRNA | Dharmacon | J-180826–09; J-180826–16; J-180826–17 | |
| Sequence-based reagent | AdipoR2 siRNA | Dharmacon | J-007801 | |
| Sequence-based reagent | NT siRNA | Dharmacon | D-001810–10 | Non-target control |
| Sequence-based reagent | FADS2 siRNA | Dharmacon | J-008211–09 | |
| Sequence-based reagent | PPIB siRNA | Dharmacon | D-001820–10 | |
| Commercial assay or kit | FITC Annexin V Apoptosis Detection Kit I | BD Bioscience | Cat No 556547 | |
| Commercial assay or kit | RevertAid H Minus First Strand cDNA Synthesis Kit | ThermoFisher | K1631 | |
| Commercial assay or kit | High Capacity cDNA Reverse Transcription Kit | Applied Biosystem | 10400745 | |
| Commercial assay or kit | Hot FIREPol EvaGreen qPCR Supermix | Solis Biodyne | 08-36-00001 | |
| Commercial assay or kit | Viromer Blue | Lipocalyx | VB-01LB-01 | |
| Chemical compound, drug | Palmitic acid | Sigma-Aldrich | P0500 | |
| Chemical compound, drug | Linoleic acid | Sigma-Aldrich | L1376 | |
| Chemical compound, drug | Eicosapentaenoic acid | Sigma-Aldrich | E2011 | |
| Chemical compound, drug | [9,10-$^3$H(N)]-Palmitic Acid | Perkin Elmer | NET043001MC | |
| Chemical compound, drug | Linoleic acid-[$^{13}$C18] | IsoSciences | S14495-1.0 | IsoSciences provided a stock dissolved in DMSO |
| Chemical compound, drug | BODIPY 500/510 C1,C12 | Invitrogen | D3823 | |

### C. elegans strains and cultivation

The wild-type *C. elegans* reference strain N2 and the mutant alleles studied are available from the *C. elegans* Genetics Center (CGC; MN; USA). The *pfat-7::GFP (rtIs30)* carrying strain HA1842 was a kind gift from Amy Walker (*Walker et al., 2011*), and its quantification was performed as previously described (*Svensk et al., 2013*). The *C. elegans* strains maintenance and experiments were performed at 20°C using the *E. coli* strain OP50 as food source, which was maintained on LB plates kept at 4°C (re-streaked every 6–8 weeks) and single colonies were picked for overnight cultivation at 37°C in LB medium then used to seed NGM plates (*Sulston and Methods, 1988*); new LB plates were streaked every 3–4 months from OP50 stocks kept frozen at −80°C. NGM plates containing 20 mM glucose were prepared using stock solution of 1 M glucose that was filter sterilized then added to cooled NGM after autoclaving. For experiments involving the mutant *fat-2(wa17)*, worms were maintained using the *nt1* balancer which balances the right half of LG IV; gravid hermaphrodites were bleached and the synchronized non-balanced progeny were scored for the growth.

### Screen for suppressors of SFA intolerance and whole genome sequencing

*paqr-2(tm3410)*, *iglr-2(et34)* or *paqr-2(tm3410) mdt-15(et14)* worms were mutagenized for 4 hr by incubation in the presence of 0.05 M ethyl methane sulfonate (EMS) according to the standard protocol (*Sulston and Methods, 1988*). The worms were then washed and placed on a culture dish. Two hours later, vigorous hermaphrodite L4 animals were transferred to new culture plates. Five days later, F1 progeny were bleached, washed and their eggs allowed to hatch overnight in M9 (22 mM KH2PO4, 42 mM Na2HPO4, 85.5 mM NaCl and 1 mM MgSO4). The resulting L1 larvae were transferred to new plates containing 20 mM glucose then screened 72 hr later for fertile adults, which were picked to new plates for further analysis.

The isolated suppressor alleles were outcrossed 4-to-6 times prior to whole genome sequencing (see below), and 10 times prior to their phenotypic characterization or use in the experiments presented here. The genomes of suppressor mutants were sequenced to a depth of 25-40x as previously described (*Sarin et al., 2008*). Differences between the reference N2 genome and that of the mutants were sorted by criteria such as non-coding substitutions, termination mutations, splice-site mutations, etc. (*Bigelow et al., 2009*). For each suppressor mutant, one or two hot spots, that is small genomic area containing several mutations, were identified, which is in accordance to previous reports (*Zuryn et al., 2010*). Mutations in the hot spot that were still retained after 10 outcrosses were considered candidate suppressors and tested experimentally as described in the text.

### Pre-loading of *E. coli* with fatty acids

Stocks of 0.1 M palmitic acid (Sigma) dissolved in ethanol, 3.1 M EPA (Sigma) or 3.24 M LA (Sigma) were diluted in LB media to final concentrations of 0.25–2 mM, inoculated with OP50 bacteria, then shaken overnight at 37°C. The bacteria were then washed twice with M9 to remove fatty acids and growth media, diluted to equal $OD_{600}$, concentrated 10X by centrifugation, dissolved in M9 and seeded onto NGM plates lacking peptone (200 μl/plate). Worms were added the following day.

### Growth, tail tip scoring and other *C. elegans* assays

For length measurement studies, synchronized L1s were plated onto test plates seeded with *E. coli*, and worms were mounted then photographed 72 hr or 96 hr later (as indicated). The length of >20 worms was measured using ImageJ (*Ballweg and Ernst, 2017*; *Ginsberg et al., 1982*). Quantification of the withered tail tip phenotype was done on synchronous 1 day old adult populations, that is 72 hr post L1 (n ≥ 100) (*Svensk et al., 2013*). Other assays starting with 1 day old adults have also previously been described in details: total brood size (n = 12) (*Svensson et al., 2011*), lifespan (n = 5) (*Svensson et al., 2011*), defecation period (n = 5; average interval between five defecation was determined for each worm) (*Liu and Thomas, 1994*), pharyngeal pumping rate (n = 10, each monitored for 20 s) (*Axäng et al., 2007*), speed (n = 10) (*Tajsharghi et al., 2005*), survival at 30°C and 37°C (n = 20) (*Zevian and Yanowitz, 2014*) and expression levels of *zcIs4(phsp-4::GFP)* (*Rauthan et al., 2013*).

## Oxidative stress assay

Oxidative stress was analyzed in N2 and *fld-1* worms using CL2166 (dvIs19 [(pAF15)gst-4p::GFP:: NLS] *III* (*Link and Johnson, 2002*). Synchronized *gst-4::GFP* and *fld-1*; *gst-4::GFP* worms were spotted on control plates and 40 mM LA plates. After 72 hr of incubation at 20°C, worms were mounted on agarose padded slides. Images were taken using Carl Zeiss high magnification microscope. Identical settings and exposure time was used for all the conditions. Quantitative measurements of the GFP intensity was performed on synchronized 1 day old adults using Image J (n ≥ 20).

## Fluorescence recovery after photobleaching (FRAP) in *C. elegans* and HEK293 cells

FRAP experiments in *C. elegans* were carried out using a membrane-associated prenylated GFP reporter expressed in intestinal cells, as previously described and using a Zeiss LSM700inv laser scanning confocal microscope with a 40X water immersion objective (*Devkota et al., 2017*; *Svensk et al., 2016a*). Briefly, the GFP-positive membranes were photobleached over a circular area (seven pixel radius) using 20 iterations of the 488 nm laser with 50% laser power transmission. Images were collected at a 12-bit intensity resolution over 256 × 256 pixels (digital zoom 4X) using a pixel dwell time of 1.58 μsec, and were all acquired under identical settings. For FRAP in mammalian cells, HEK293 cells were stained with BODIPY 500/510 $C_1$, $C_{12}$ (4,4-Difluoro-5-Methyl-4-Bora-3a,4a-Diaza-*s*-Indacene-3-Dodecanoic Acid) (Invitrogen) at 2 μg/ml in PBS for 10 min at 37°C (*Devkota et al., 2017*). FRAP images were acquired with an LSM880 confocal microscope equipped with a live cell chamber (set at 37°C and 5% $CO_2$) and ZEN software (Zeiss) with a 40X water immersion objective. Cells were excited with a 488 nm laser and the emission between 493 and 589 nm recorded. Images were acquired with 16 bits image depth and 256 × 256 resolution using a pixel dwell of ~1.34 μs. At least ten (n ≥ 10) pre-bleaching images were collected and then the region of interest was beached with 50% of laser power. The recovery of fluorescence was traced for 25 s. Fluorescence recovery and $T_{half}$ were calculated as previously described (*Svensk et al., 2016a*). Videos of FRAP experiments are provided as supplementary material (Figure 4 –videos 1-3).

## Western blot

Crowded plates of various *C. elegans* strains were harvested and washed twice in M9 and then lysed using the following lysis buffer: 25 mM Tris (pH 7.5), 300 mM Nacl, 1% triton X-100 and 1X protease inhibitor. Total of 100 μg and 25 μg (for transgenic worms) of protein was loaded in each lane. For detection of the FLD-1 protein, rabbit antiserum was generated by immunizing a rabbit against the peptide TQVGDVESGPLRTQ corresponding to C terminal amino acids (GeneScript). The nitrocellulose membranes (GE Healthcare) were blocked with 5% skimmed milk diluted in TBST. Antibody dilutions (primary antibody: (1:500) and secondary antibody: (1:2500)) and washes were carried out in TBST. Detection was performed using an ECL detection kit (Millipore), as per the manufacturer's instructions.

## Plasmids

### Wild-type *fld-1* rescue construct

The P*fld-1::fld-1* rescue construct was created using the primers 5'- agattttgaggcttttctgtggg −3' and 5'- gcagctgcctcatttgttga −3' to amplify the *fld-1* gene and 2 kb of upstream regulatory sequence using N2 worm genomic DNA as template. The resulting 6.7 kb PCR product was cloned into the pCR XL-TOPO vector using TOPO XL PCR cloning kit (Invitrogen). The assembled plasmid was injected into N2 worms at 10 ng/μl together with 3 ng/μl pPD118.33 (*Pmyo-2::GFP*), a gift from Andrew Fire (Addgene plasmid # 1596); (*Davis et al., 2008*).

### *fld-1* translational GFP reporter

The *fld-1* translational GFP reporter was generated with a Gibson assembly cloning kit (NEB) by assembly of the following 2 DNA fragments: (1) The *fld-1* promoter and *fld-1* coding sequence was amplified from the *fld-1* rescuing plasmid with the following primers: 5'-ggcatggatgaactatacaaata-gactcaccccgccgctcccccaattgt-3' and 5'- aacagtttgtgtcgcgtcttctgttgttccatgaatcgtttgcactccga-3', (2) GFP was amplified from the plasmid P*paqr-2:paqr-2:GFP* (*Svensson et al., 2011*) using the primers 5'- cagaagacgcgacacaaactgttatgagtaaaggagaagaact-3' and tgtaatcc cagcagctgttacaa-3'. The resulting P*fld-1::FLD-1::GFP* plasmid was injected into N2 worms at 10 ng/μ

l together with the plasmid 30 ng/μl *PRF4,* which carries the dominant *rol-6(su1006)* marker (*Mello et al., 1991*).

### Pelt-3::FLD-1::GFP

The *fld-1* hypodermis-specific transgene was constructed using a Gibson assembly cloning kit (NEB) with the following DNA fragments: (1) 2 kb upstream regulatory sequences from *elt-3* amplified from N2 genomic DNA using the primers 5'- cgccagtgtgctggaattcgccctttgtgacacgttgtttcacggtcatc-3' and 5'- tcaacagctccgccagctgccgcatccgagcttgctgagatggctggaca-3' and (2) the *fld-1* coding sequence with GFP was amplified from the plasmid *fld-1* translational GFP reporter using primers 5'-atgcggcagctggcggagct-3' and 5'- aagggcgaattccagcacac-3'. The assembled plasmid was injected into N2 worms at 10 ng/μl together with 3 ng/μl *pPD118.33*.

### Pges-1::FLD-1::GFP

Intestine-specific transgene was constructed using Gibson assembly cloning kit (NEB) by assembly of the following 2 DNA fragments: (1) 2 kb upstream regulatory sequences from *ges-1* was amplified from N2 genomic DNA using the primers 5'- ccgccagtgtgctggaattcgcccttaatattctaagcttaatgaagttt-3' and 5'- tcaacagctccgccagctgccgcatctgaattcaaagataagatatgtaa-3', (2) The *fld-1* coding sequence with GFP was amplified from the plasmid *fld-1* translational GFP reporter using primers 5'- atgcgg-cagctggcggagct-3' and 5'- aagggcgaattccagcacac-3'. The assembled plasmid was injected into N2 worms at 10 ng/μl together with 3 ng/μl *pPD118.33*.

## C. elegans and HEK293 lipidomics

For worm lipidomics, samples were composed of synchronized L4 larvae (one 9 cm diameter plate/sample; each treatment/genotype was prepared in five independently grown replicates) grown overnight on OP50-seeded NGM, NGM containing 20 mM glucose or plates lacking peptone but seeded with fatty acid-supplemented bacteria. Worms were washed 3 times with M9, pelleted and stored at −80°C until analysis. For HEK293 lipidomics, cells (prepared in at least three independent replicates) were cultivated in serum-free media with or without fatty acids for 24 hr prior to harvesting using TrypLE Express (Gibco). For lipid extraction, the pellet was sonicated for 10 min in methanol and then extracted according to published methods (*Löfgren et al., 2012*). Internal standards were added during the extraction. Lipid extracts were evaporated and reconstituted in chloroform:methanol [1:2] with 5 mM ammonium acetate. This solution was infused directly (shotgun approach) into a QTRAP 5500 mass spectrophotometer (Sciex, Toronto, Canada) equipped with a Nanomate Triversa (Advion Bioscience, Ithaca, NY) as described previously (*Jung et al., 2011*). Phospholipids were measured using multiple precursor ion scanning (*Ejsing et al., 2009*; *Ekroos et al., 2003*). Ceramides from HEK293 cells were measured using ultra performance liquid chromatography coupled to tandem mass spectrometry according to previous publication (*Amrutkar et al., 2015*). Free cholesterol from HEK293 cells was quantified using straight phase HPLC coupled to ELS detection according to previous publication (*Homan and Anderson, 1998*). The data were evaluated using the LipidView software (Sciex, Toronto, Canada). The complete lipidomics dataset is provided as source data files associated with the figures.

## Expression profiling of TLCD1 and TLCD2 in mouse

Six isoflurane-anesthetized C57BL/6 mice (Charles River) were terminated (14–16 $_{P.M.}$) and organs were collected, weighed and snap frozen in liquid nitrogen for further storage at −80°C. Total RNA extraction was done using TRIzol LS (ThermoFisher Scientific) on all tissues along with a RNeasy Mini Kit or RNeasy Fibrous Mini Kit for fiber-rich tissues (heart and skeletal muscle) according to manufacturer's instructions (Qiagen). Quantification was done using a NanoDrop spectrophotometer (ND-1000; ThermoFisher Scientific) with quality cut-off set at >1.8 for both $A_{260}/A_{280}$ and $A_{260}/A_{230}$ before synthesizing cDNA using a High Capacity cDNA Reverse Transcription Kit (Applied Biosystem) with random hexamers. qPCR was executed in triplicates on a QuantStudio7 Flex Real-Time PCR System (ThermoFisher Scientific) using a TaqMan Gene Expression Master Mix (ThermoFisher Scientific) and TaqMan Gene Expression Assays for all genes: TLCD1 (Mm00503356_g1), TLCD2 (Mm01240453_g1). PPIA (Mm02342430_g1) was used to normalize for RNA input variation. FAM fluorophore with MGB was used as probe. All genes were measured in separate wells in technical

triplicates but on the same plate, and with at least three independent biological samples for each tissue. The relative gene expression was calculated as $2^{(-\Delta\Delta Ct)}$ using the lowest expressing tissue as reference tissue as described elsewhere (*Livak and Schmittgen, 2001*). All mice experimental work was approved by the local ethics review committee on animal experiments (Gothenburg region).

## Cultivation of HEK293

HEK293 (identify verified by STR profiling and tested free of mycoplasma) were grown in DMEM containing glucose 1 g/l, pyruvate and GlutaMAX and supplemented with 10% fetal bovine serum, 1% non-essential amino acids, HEPES 10 mM and 1% penicillin and streptomycin (all from Life Technologies) at 37°C in a water humidified 5% $CO_2$ incubator. Cells were sub-cultured twice a week at 90% confluence. Cells were cultivated on treated plastic flask and multi-dish plates (Nunc). For FRAP experiments, HEK293 were seeded in glass bottom dishes (Ibidi) pre-coated with 0.1% porcine gelatin (Sigma). Millicell EZ slides (Millipore) were used for Nile red staining.

## siRNA in HEK293 cells

The following pre-designed siRNAs were purchased from Dharmacon: AdipoR2 J-007801–10, Nontarget D-001810–10, PPIB D-001820–10, TLCD1 J-015483–10, TLCD2 J-180826–09 (#1) J-180826–16 (#2) and J-180826–17 (#3). Transfection of 25 nM siRNA was performed in complete media using Viromer Blue according to the manufacturer's instructions 1X (Lipocalyx). Knockdown gene expression was verified 48 hr and/or 72 hr after transfection.

## Quantitative PCR in HEK293 cells

Total cellular RNA was isolated using RNeasy Kit according to the manufacturer's instructions (Qiagen) and quantified using a NanoDrop spectrophotometer (ND-1000; Thermo Scientific). cDNA was obtained using a High Capacity cDNA Reverse Transcription Kit (Applied Biosystem) or a RevertAid H Minus First Strand cDNA Synthesis Kit with random hexamers. qPCR were performed with a CFX Connect thermal cycler (Bio Rad) using Hot FIREpol EvaGreen qPCR SuperMix (Solis Biodyne) and standard primers. Samples were measured as triplicates. The relative expression of each gene was calculated according to the ΔΔCT method (*Livak and Schmittgen, 2001*). Expression of the housekeeping gene PPIA was used to normalize for variations in RNA input. Primers used were: AdipoR2-For (TCATCTGTGTGCTGGGCATT) and -Rev (CTATCTGCCCTATGGTGGCG), FADS-1-For (TGGCTAGTGATCGACCGTAA) and –Rev (GGCCCTTGTTGATGTGGAAG), FADS-2-For (GGGCCGTCAGCTACTACATC) and –Rev (ACAAACCAGTGGCTCTCCAG), FADS-3-For (AATCTGGCGTACATGCTGGT) and –Rev (AAGAGGTGGTGCTCGATCTG), PPIA-For (GTCTCCTTTGAGCTGTTTGCAG) and -Rev (GGACAAGATGCCAGGACCC), PPIB-For (AGATGTAGGCCGGGGTGATCT) and –Rev (GCTCTTTCCTCCTGTGCCAT), SCD-For (TTCGTTGCCACTTTCTTGCG) and –Rev (TGGTGGTAGTTGTGGAAGCC), TLCD1-For (TGGCACAACCTGCTCGTC) and –Rev (CACGATGTCCACCGTATCGT), TLCD2-For (TGGCTGTGTCTGTGGGTTAC) and –Rev (AGCAGGCAGAGTTCAGTTCC). Changes in gene expression were measured in $n \geq 3$ biological independent experiment, including internal technical triplicates.

## HEK293 fatty acid treatment

PA, LA, EPA and DHA were dissolved/diluted in sterile DMSO (Sigma) then mixed with fatty acid-free BSA (Sigma) in serum-free medium for 15 min at room temperature. LA-[$^{13}C_{18}$] in DMSO solution was obtained from IsoSciences and used at 50 µM. The molecular ratio of BSA to PA was 1 to 5.3 (except in the experiment using 200 µM PA in which case the ratio was 1 to 2.65). Cells were then cultivated in this serum-free media containing the fatty acids for 24 hr prior to analysis, unless stated otherwise.

## Annexin V staining and flow cytometry

Cells were detached with TrypLE Express (Gibco), washed and resuspended in Annexin V binding buffer (BD Bioscience). Then, cells were stained with FITC-Annexin V and 7-ADD according manufacturer's instructions (BD Bioscience) and analyzed in a FACS Canto II (BD Bioscience) flow cytometer. Data were analyzed with FlowJo X. Early apoptotic cells were defined by Annexin V$^+$ and 7-ADD$^-$. Apoptotic cells were measured in three independent experiments.

## Palmitic acid uptake

Cells were treated with [9,10-3H(N)]-Palmitic Acid (Perkin Elmer) at a specific activity of 0.5 µCi and 100 µM cold PA in 0.5% BSA. After 10, 30 or 60 min of incubation, the cells were washed with PBS, resuspended and the activity measured with a TRI-CARB 4810TR 110 V Liquid Scintillation Counter (Perkin Elmer). Fatty acid uptake inhibitor phloretin (Sigma) was dissolved in DMSO. When phlotetin was used, cells were pre-treated for 15 min and the concentration kept constant at 200 µM.

## Nile red staining

Staining was carried out in fixed cells (3.7% formaldehyde, 20 min). The dye was added directly to the fixed cells to a final concentration of 100 ng/ml and incubated for 30 min. The dye was washed out using PBS prior to microscopy. Hoescht3458 was added to visualize cell nuclei. Fluorescence signal was captured with a Zeiss Axio Scope.A1 microscope and AxioVision 4.7.2 software. Images were automatically quantified with Image J software.

## Statistics

Except for the scoring of the worm tails, error bars show the standard error of the mean, and $t$-tests were used to identify significant differences between treatments. Error bars for the frequency of the tail tip defect show the 95% confidence interval and significant differences were determined using $Z$-tests. $t$-tests were used to determine significance in FRAP experiments, and when comparing lipidomics data. Asterisks are used in the figures to indicate various degrees of significance, where *: $p<0.05$; **: $p<0.01$; and ***: $p<0.001$. All quantitative experiments were performed with a minimum of three independent biological replicates (see individual Materials and methods), meaning that populations of worms or cells were independently grown and analysed. All quantitative experiments, with the exception of some lipidomics experiments, were also entirely performed at least twice with similar results. The exceptions were single runs of the following lipidomics experiments (each run included at least four independent biological replicates): LA-[$^{13}C_{18}$] experiment in *Figure 4H–I* and *Figure 4—figure supplement 3K-L*.

## Acknowledgments

We thank the Center for Cellular Imaging at the University of Gothenburg for microscopy support. Some *C. elegans* strains were provided by the *Caenorhabditis elegans* Genetics Center, which is funded by the NIH Office of Research Infrastructure Programs (P40 OD010440). We also thank Svenja Mende, Vilma Canfjorden and Rebecka Persson for their help with some of the experiments. Funding: Cancerfonden, Vetenskaprådet, Carl Tryggers Stiftelse, Diabetesfonden, Swedish Foundation for Strategic Research, Kungliga Vetenskaps-och Vitterhets-Samhället i Göteborg, Lundgrens Stiftelse and Tore Nilssons Stiftelse. There are no competing interests. All data are available in the manuscript or the supplementary materials.

## Additional information

### Competing interests

Henrik Palmgren: affiliated with AstraZeneca. The author has no other competing interests to declare. The other authors declare that no competing interests exist.

### Funding

| Funder | Author |
| --- | --- |
| Kungl. Vetenskaps- och Vitterhets-Samhället i Göteborg | Mario Ruiz<br>Rakesh Bodhicharla<br>Emma Svensk<br>Ranjan Devkota<br>Marc Pilon |
| Stiftelserna Wilhelm och Martina Lundgrens | Mario Ruiz<br>Ranjan Devkota<br>Rakesh Bodhicharla |

| Tore Nilsons Stiftelse för Medicinsk Forskning | Mario Ruiz |
| --- | --- |
| Vetenskapsrådet | Marc Pilon |
| Cancerfonden | Marc Pilon |
| Carl Tryggers Stiftelse för Vetenskaplig Forskning | Marc Pilon |
| Diabetesfonden | Marc Pilon |
| Swedish Foundation for Strategic Research | Marc Pilon |

The funders had no role in study design, data collection and interpretation, or the decision to submit the work for publication.

## Author contributions

Mario Ruiz, Rakesh Bodhicharla, Emma Svensk, Ranjan Devkota, Conceptualization, Investigation, Visualization, Writing—review and editing; Kiran Busayavalasa, Henrik Palmgren, Marcus Ståhlman, Conceptualization, Investigation, Visualization; Jan Boren, Conceptualization, Supervision, Funding acquisition, Investigation, Visualization, Project administration, Writing—review and editing; Marc Pilon, Conceptualization, Supervision, Funding acquisition, Investigation, Visualization, Writing—original draft, Project administration, Writing—review and editing

## Author ORCIDs

Marcus Ståhlman https://orcid.org/0000-0002-4202-0339
Marc Pilon http://orcid.org/0000-0003-3919-2882

## Decision letter and Author response

Decision letter https://doi.org/10.7554/eLife.40686.046
Author response https://doi.org/10.7554/eLife.40686.047

---

# Additional files

## Supplementary files

• Transparent reporting form
DOI: https://doi.org/10.7554/eLife.40686.043

## Data availability

All data generated or analysed during this study are included in the manuscript and supporting files. The complete lipidomics dataset is provided as source data files associated with the figures.

---

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
