## [Decision Letter]

[**Editorial note:** This article has been through an editorial process in which the authors decide how to respond to the issues raised during peer review. The Reviewing Editor's assessment is that all the issues have been addressed.]

Thank you for submitting your article "Membrane Fluidity is Regulated by the *C. elegans* Transmembrane Protein FLD-1 and its Human Homologs TLCD1/2" for consideration by *eLife*. Your article has been reviewed by three peer reviewers, and the evaluation has been overseen by a guest Reviewing Editor and a Senior Editor. The following individual involved in review of your submission has agreed to reveal their identity: Robert Ernst (Reviewer #2). Two other reviewers remain anonymous.

All three reviewers have provided detailed and thoughtful reviews, and we have included the separate reviews below for your consideration.

Overall the reviewers were in broad agreement about the interest of the work and the high quality and compelling nature of the genetic analysis. They also identified a number of weaknesses that they felt needed to be addressed prior to publication. In particular the evidence for the localisation of FLD-1 and its mammalian orthologs, and the effects of the mutations on the levels of cholesterol needs to be addressed more clearly. In addition they felt that your description of what the data show and the citation of the literature need to be more accurate, and some of the statistical analysis needs to be improved. These seem the key points, but the more minor points they make also seem valid and hopefully most can be addressed by changes to the text.

If you can address all of the reviewers' points, and provide a point-by-point description of your revisions, then I would be happy to consult them again to ask if the paper is now of acceptable quality.

Separate reviews (please respond to each point):

*Reviewer #1:*

The authors conduct an extensive genetic approach in worms to characterize the FLD-1 protein and then performed experiments with mammalian cells to study the FLD-1 homologues, TLCD1 and TLCD2. Together with careful lipidomics data and measurements of plasma membrane dynamics by FRAP experiment, this study suggests that these mysterious transmembrane proteins prevent excessive incorporation of polyunsaturated acyl chains into phospholipids, notably PE and PC. When these proteins are deleted or silenced, PUFAs incorporation is increased, which can protect cells from the rigidification effect of lipid saturation (e.g. under the presence of glucose or at low temperature). Although the exact molecular activities of TLCD1/2 and FLD-1 remains unknown, this study is very original and the data are compelling. As this family exists in many organisms, the work should interest a wide audience.

Specific points:

1) Plasma membrane localization of FLD1. This localization is inferred from confocal images using a GFP construct (Figures 2F and Figure 1—figure supplement 4). I notice that the cells shown in Figure 2F have a prominent nucleus, making the cytoplasm volume minimal. Therefore, I'm not entirely convinced that the protein is at the plasma membrane. Wouldn't a protein present in the cortical ER give a similar localization pattern? This is important because, functionally, the localization of FLD1 at the plasma membrane is puzzling considering that lipid synthesis and remodeling is believed to occur mostly in internal membranes (notably ER). How can FLD1 affects fatty acid incorporation into phospholipids if the protein is not in the same membrane as lysophospholipid acyl transferases? See also my remark on the mammalian proteins.

2) The genetic characterization in Figures 2 and 3 is extensive and compelling, showing a clear rescue effect of various *fld-1* mutants on the defects induced by temperature or glucose on the growth and morphology of worm mutants for the PAQR-2 and/or IGLR-2 proteins. The genetic interactions tested in Figures 2D-F and 3E suggest that FLD-1 acts by a pathway different from the previously described pathway, which relies on the overexpression of a FA desaturase. Instead, they suggest a genetic interaction with the desaturases *fat-6* or *fat-7* (please comment on the lack of interaction with *fat-5*; Figure 3E). How about acyl transferases e.g. MBOA-6?

3) Related to the previous point. At least in text or perhaps through further genetic studies, the authors should try to connect their work with recent studies on the families of acyltransferases responsible for the incorporation of fatty acids with defined level of unsaturation (e.g. PMID:28578316 and others studies by the Shimizu group), as well as on recent attempts to better characterize the molecular properties of polyunsaturated phospholipids in membranes (e.g. Pinot et al., 2014 for endocytosis).

4) The lipidomic analysis on the acyl chain profile performed on phosphatidylethanolamine is also compelling. Under glucose, the *paqr-2* mutant accumulates saturated PE at the expense of polyunsaturated PE, an effect that is inverted by the further introduction of the *fld-1* mutation, notably in combination with mutation of the *mdt-15* gene. The data under conditions of a saturated diet (Figure 3—figure supplement 1B-D) are also impressive. The authors, however, might mention that the plasma membrane accounts for only a minor part of the total cellular membranes. Therefore, it is unlikely that the plasma membrane is the only one to be affected in its acyl chain profile. The lipidomic analysis is performed on cells or tissues, not on plasma membrane, which is technically much more demanding.

5) For mammalian cells, would it be possible to further study the sub-localization of the FLD-1 homologues (TLCD1 and TLCD2).

Minor Comments:

6) I am not sure I understood the experiments shown in Figure 4—figure supplement 2O and the associated conclusion. Is there an effect of silencing TLCD1 or TLCD2 on desaturases expression?

Additional data files and statistical comments:

The running title is: TLCD1/2 limit LCPUFAs. It should be TLCD1/2 limit LCPUFA incorporation.

*Reviewer #2:*

This genetic study describes the identification of an important modulator of the membrane lipid composition in *C. elegans* and human cells. Building on previous work by the same group, in which the plasma membrane-localized PAQR-2 and IGLR-2 proteins have been identified as an important modulators of fatty acid desaturases, the authors used genetic screens in *paqr-2* and *iglr-2* mutant backgrounds to identify other regulatory factors. The genetic evidence for *fld-1* being an important suppressor of *paqr-2* mutant variants employing a distinct mechanism is very convincing. Furthermore, the authors present data that *fld-1* is ubiquitously expressed in plasma membranes and that *fld-1* mutant alleles suppress a series of membrane-related defects observed in *paqr-2* cells, but do not seem to affect membrane properties and membrane-related phenotypes in a wildtype background. The phenotypes include membrane fluidity (as tested by FRAP experiments), worm length, and the acyl chain composition of membrane lipid. Lipidomic analysis provides strong evidence that *fld-1* limits the level of long chain polyunsaturated membrane lipids and that *fld-1* mutants increase it.

Knockdown experiments of the mammalian TLCD1 and TLCD2 proteins identify similar effects on the membrane lipid composition and membrane fluidity in HEK293 cells stressed by palmitic acid provided with the medium. Pulse-labeling experiment suggest that the production/formation of polyunsaturated acyl chain-containing lipids is increased in *fld-1* mutants thereby leading to the conclusion that *fld-1* is a key regulator of polyunsaturated lipid homeostasis.

Without any doubt, this manuscript identifies *fld-1* and TLCD1/2 as important contributors to membrane homeostasis in *C. elegans* and human cells. The technical quality and presentation of the data is very good. The conclusions of the manuscript are justified and provide an important, new link for a better understanding of membrane lipids in health and disease. After a revision of the manuscript based on the points discussed below, I can recommend this manuscript for publication in *eLife*.

Strength/Weaknesses:

Even though the number of weaknesses outnumbers the number of strengths, I consider this manuscript as an excellent study with potentially very high impact on a much broader scientific community in the future.

Strength 1: This manuscript represents a thorough genetic dissection of the pathways involved with membrane homeostasis paired with an extensive phenotypic characterization including lipidomic analyses.

Strength 2: The technical quality of the data is very high and support the conclusions of the authors.

Strength 3: The authors provide compelling and carefully controlled data from two distinct model systems: *C. elegans* and human cells.

Weakness 1: The literature referenced in the Introduction and Discussion does often not suit the purpose. Many papers referred to in the text report correlative observations, but are cited in way to make a statement. Only few studies mentioned in the text provide the original, clear-cut, biochemical or biophysical studies for the point being made. In some instances, there seems to be a discrepancy between the cited reference and the statement made in the text. This must be improved.

Examples:

Weijers, 2016: The role of membrane 'flexibility' on microcirculation does not become clear to the reader. The term 'flexibility' lacks a clear physical definition.

Weijers, 2012 is a review article with similar content as Weijers, 2016, while other highly influential references such as (e.g. PMID: 22385956) or original literature linking the membrane lipid composition to diabetes are missing.

Taraboletti et al., 1989 and Sherbet, 1989 do not justify the statement that defects in membrane are hallmarks of cancer. Taraboletti et al., 1989 provides only a correlative argument, while the Abstract of Sherbet, 1989 even states 'Steady-state fluorescence polarization […] does not suggest marked differences in bulk 'fluidity' of metastatic variant'.

Fecchio, Palazzi and de Laureto, 2018 primarily discusses the folding of α-synuclein in the presence of high concentration of free polyunsaturated fatty acids and in the absence of cellular membranes. This is not a suitable model to imply functions of PUFAs in neurodegenerative diseases. The role of α-synuclein in interacting with cellular membranes and regulating vesicular traffic is not being discussed at all in this context.

Other recent studies, which suggest important roles in specific polyunsaturated lipids in membrane deformation (PMID:25104391), domain stability (PMID: 27119640), membrane fission events (PMID: 29543154). Potential players important for mediated the regulated production/conversion of membrane lipids in *fld-1* cells are not discussed (PMID: 18782225; PMID: 18094042; PMID: 7174678). I suggest that the authors carefully revisit their list of references used in the Introduction and the Discussion.

Weakness 2: The concept of membrane fluidity is highly prevalent in biology and serves as a wonderful descriptor of membrane biophysics, which is also intuitively understood by a broader readership. However, I do not find any clear evidence in this manuscript (nor in the cited references) that membrane fluidity is indeed the critical factor in the context of the observed phenotypes. The role of lipids containing polyunsaturated fatty acids has gained increasing attention in the past years (see below) and the authors could do a better job in highlighting these developments in light of their data.

The authors should clearly indicate that membrane fluidity is used as a readily available descriptor of a membrane property. A separate paragraph should discuss that also other membrane properties are greatly affected by lipids with polyunsaturated fatty acid chains and that it is not clear, which of these membrane properties are most relevant for the observed phenotypes observed in genetic experiments.

Weakness 3: The important role of FLD-1 and TLCD1/2 is being established, but the potential targets of FLD-1 and TLCD1/2 are neither discussed, nor is there any speculation on how a signal from FLD-1 or TLCD1/2 might be transmitted to regulate the production/formation of very long chain polyunsaturated lipids. The authors should provide some more insight in how the communication between the plasma membrane bound proteins and their effector functions is being established. A direct regulation of FLD-1 by PAQR-2 is indicated in Figure 6, but I cannot find evidence that supports this speculation. Do the authors suggest a direct regulation or rather an indirect one via the fluidity (or other properties) of the plasma membrane?

How can a plasma membrane embedded protein send a signal that affects the incorporation of polyunsaturated fatty acids into lipids? Does it signal go to the ER, to the nucleus, or to the plasma membrane? Is there any indication for a transcriptional response or for a signaling molecule/second messenger/post-translational modification that might transmit the signal to other proteins involved in establishing the fatty acyl chain composition of membrane lipids? Does, for example, knockdown of TLCD1/2 lower the activity of the unfolded protein response in response to palmitate? This would indicate that the lipid composition in the ER membrane is affected FLD-1 mediated regulation and not only the plasma membrane.

What is the connection to cholesterol?

An important factor contributing to membrane fluidity and membrane phase behavior of cellular membranes in mammals is cholesterol. Especially because cholesterol has key functions in organizing the plasma membrane of mammalian cells (but not *C. elegans*), and because SREBP proteins are central regulators of many aspects of lipid metabolism, it would be important to see if and how TLCD1/2 affects the cholesterol level and cholesterol signaling in these cells.

Related to the FRAP experiments

For HEK cells and *C. elegans* different approaches have been undertaken. Does incubation at 37 °C of the BODIPY 500/510 for 10 min result in an uptake of the dye to inner membranes? In order words: do the FRAP data refer to FRAP in the plasma membrane or is it due to internal membranes? The authors should provide a confocal image that shows incorporation of the dye in the plasma membrane and no strong labeling of intracellular membranes. Furthermore, the authors should use a GPI-anchored protein to measure the effect of FLD-1 and/or TLCD1/2 on proteins on the extracellular leaflet of the plasma membrane.

Other points:

General: The number of individual data points should be provided with the experiments and with the appropriate statistical tests.

Subsection “*fld-1* is ubiquitously expressed in plasma membranes”: The TLC (TRAM, LAG1,and CLN8) domain should be introduced better – otherwise it may remain unclear to many readers what TRAM, LAG1, and CLN8 stand for.

Subsection “*fld-1* acts in a pathway distinct from other *paqr-2* suppressors”: 'PA' is used as an abbreviation for palmitic acid, but introduced only later in the paragraph. Moreover, the abbreviation may be misleading, as the paper refers to phosphatidylethanolamine as PE, and PA might be misunderstood as phosphatidic acid. Maybe the authors find a less confusing way to refer to palmitic acid?

Subsection “*fld-1* acts in a pathway distinct from other *paqr-2* suppressors”: the *mdt-15* and *cept-1* mutant worms are mentioned without further explanation of their putative functions. This might be confusing for a reader not familiar with *C. elegans*.

– Subsection “Mammalian TLCD1 and TLCD2 are regulators of membrane composition and fluidity”, second paragraph and Figure 4—figure supplement 2OFigure: The expression level of SCD is substantially reduced, but this observation is not discussed in the manuscript (only in the figure legend). Does this indicate that there are substantial changes in the lipid composition of the endoplasmic reticulum under these conditions, which lead to a distinct regulation of SCD?

The notion that 'as little as 1 µM of EPA or DHA […] prevents membrane rigidification by 400 µM PA' should be avoided. It does not make sense to compare these concentrations, because the uptake, metabolism, incorporation in different lipids, their subcellular localization, and turnover might differ substantially between these fatty acids. Just comparing the concentration in the media does not make sense.

Subsection “Mammalian TLCD1 and TLCD2 are regulators of membrane composition and fluidity”, last paragraph: (as above) Comparing the concentration of LCPUFA necessary to protect cells from 200 µM PA is not relevant, given the complexity of the metabolic steps involved.

Discussion, first paragraph: (similar as above): The expression 'fifty times more potent' is misleading and should be omitted. The cited reference (Yang et al., 2011) does not warrant equal metabolism of LCPUFAs and oleic acid in cells. This statement would ignore, that different fatty acids have remarkable distinct abundancies in the cellular membrane. The statement does not define in what respect LCPUFAs are more potent? How is it being measured? Of note, the effect of different lipids on collective bilayer properties are not linear. For these reasons, the authors should avoid a quantitative comparison of the effects by different fatty acids.

Discussion, second paragraph: The authors list a number of processes that seem to be affected by the membrane lipid composition. However, the true underlying mechanism remain entirely unexplored in these studies. Also, membrane microdomains are mentioned here for the first time in the manuscript. This of course has many implications on the interpretation of the FRAP experiments (see below). I would recommend not to go into a discussion on the modulatory role of the membrane composition, but instead focus more on the discussion how FLD-1 and/or TLCD1/2 might excerpt their modulatory function.

Discussion, third paragraph: It is unclear to me, how the presented data suggest a direct, regulatory role of PAQR-2 and IGLR-2 on FLD-1. The proteins might act independently. However, if PAQR-2 and IGLR-2 would modulate FLD-1 function – what would be the signal?

Figure 2C: The most interesting statistical test should be performed between worms with the *fld-1* allele and without it.

Additional data files and statistical comments:

As mentioned in the previous comments, it would help if the number of experiments (n) would be provided with the experiments.

*Reviewer #3:*

We have the read the manuscript 'Membrane fluidity is regulated by the *C. elegans* transmembrane protein FLD-1 and its human homologs TLCD1/2' by Ruiz et al. In this study the authors have performed a suppressor screen for *paqr* (adiponectin receptor homolog) and *iglr* (adiponectin interacting protein) mutants that show increased membrane rigidity at low temperature or when grown in presented on high glucose. They identified several alleles of this gene were capable of suppressing the growth and morphological defects observed in the two mutants. They show that *fld-1* mutants, under conditions of stress (cold/high glucose/or feeding of saturated fatty acid) continue to favor a membrane composition that is slightly higher in unsaturated fatty acids thus increasing its fluidity. They have demonstrated similar abilities for human homologs of this protein in tissue culture studies.

Strengths of the study:

1) Suppressor screen and identification of genetic interactors of PAQR/Adiponectin receptor.

2) Detailed genetic studies of *paqr/iglr/fld-1*/and other previously isolated suppressors of *paqr/iglr*.

3) Efforts to extend the original findings in C.elegans to human homologs in tissue culture experiments.

4) Lipidomic correlation of morphological defects with increased saturated fatty acid content in membrane lipids (this is not new finding but extension of authors' findings in earlier studies).

5) General concurrence between *C. elegans* and human data.

Weakness of the study:

1) No biochemical function for FLD-1/TLCD1-2 has been elucidated in this study. The manuscript begins as a genetic interaction study and remains as such despite valiant efforts by the authors to describe the interactions using numerous experiments and extending the studies to human homologs of FLD-1.

2) The Abstract is misleading –

'We isolated *C. elegans* mutants with improved tolerance to dietary saturated fat, including eight alleles of the novel gene fld-1 that encodes ahomolog of the human TLCD1 and TLCD2 transmembrane proteins. FLD-1 is localized on plasma membranes and mutants have an excess of highly membrane-fluidizing long-chain polyunsaturated fatty acid-containing phospholipids that results in improved membrane fluidity as measured in vivo using fluorescent recovery after photobleaching (FRAP)'.

The mutants do not show notable phenotypes or deviations from normal under regular growth conditions, at least from what can be inferred from data observed in the figures and by the authors' own admission while describing the phenotypic characterization of the mutants. The mutants do not show improved tolerance to saturated fat on their own and do not have an excess of membrane-fluidizing LCPUFA ordinarily but only under stress conditions. They show improved tolerance only in the presence of other mutations.

3) Since the molecular function of FLD-1 has not been elucidated it leads to a conceptual dilemma in the working hypothesis proposed by the authors. The authors argue that FLD-1/TLCD1-2 is a plasma membrane protein that decreases the incorporation of LCPUFA to the membrane implying a function in the immediate vicinity of plasma membrane (thus cell autonomous). Yet in Figure 1—figure supplement 5C, they show transgenic expression of FLD-1 in hypodermis or intestine is sufficient to restore glucose sensitivity on *paqr-fld-1* double mutant indicating a cell non-autonomous role.

4) The data shows that in the absence of these proteins, there is an increase in incorporation of LCPUFA's within 6 hours, indicating that the deacylation-reacylation cycle (Lands cycle) may be involved. Is it possible that FLD-1/TLCD1/TLCD2 interact with membrane phospholipases or Acyl-CoA:lysopospholipid acyltransferases and modulate their function?

There are several minor corrections that need to be incorporated throughout the manuscript. A few examples are noted below:

1) Figure 3E-G the dashed line needs to be included.

2) Figure 4—figure supplement 3. Legend refers to '(R-S) Knockdown of the desaturase FADS2 reduces the membrane fluidity ofHEK293 cells cultivated in 400 μM PA but does not prevent the protective effect of TLCD2 knockdown'. These figures seem to be missing.

3) Figure 2A-B: The *fld-1(et45)* shown as single mutant but it should be *fld-1(et45); paqr-2* double mutant?

4) Subsection “*fld-1* acts in a pathway distinct from other *paqr-2* suppressors”: typo error *fld-1(et34)* it should be *fld-1(et48)*.

5) It is not clear why SDs have been used in some analyses and SEM for others?

---

## [Author Response]

Overall the reviewers were in broad agreement about the interest of the work and the high quality and compelling nature of the genetic analysis. They also identified a number of weaknesses that they felt needed to be addressed prior to publication. In particular the evidence for the localisation of FLD-1 and its mammalian orthologs, and the effects of the mutations on the levels of cholesterol needs to be addressed more clearly. In addition they felt that your description of what the data show and the citation of the literature need to be more accurate, and some of the statistical analysis needs to be improved. These seem the key points, but the more minor points they make also seem valid and hopefully most can be addressed by changes to the text.

Response to the decision letter:

1) A specific point raised in the decision letter regards the localization of the FLD-1 protein. There is no doubt that the in vivo FLD-1::GFP reporter is localized to the plasma membrane of most and perhaps even all cells in *C. elegans* at all developmental stages. We now provide an additional Figure 1—figure supplement 4A panel which shows confocal imaging of this reporter on a non-fixed living worm where it is very clear that all plasma membranes are GFP-positive while very little/no GFP expression is found in the cytoplasm. We have retained Figure 1F because it includes DAPI staining even though this requires fixation which causes some deformation of the morphology. However, we have rephrased the figure legend as follows: “Confocal image of the anterior portion of a fixed *C. elegans* L1 larva expressing the in vivo translational reporter *Pfld-1::FLD-1::GFP*. This reporter is ubiquitously expressed in the plasma membrane (green) while the DAPI staining (blue) decorates most of the cytoplasm in each cell since fixation allowed the chromosomal DNA to escape the nucleus.” The mammalian homolog TLCD1 was carefully localized to the plasma membrane in mammalian cells using immunofluorescence detection of a series of MYC-epitope tagged constructs. We now explicitly mention this as follows: “The same study localized TLCD1 to the plasma membrane of mammalian cells using a careful series of myc-epitope tagging and immunodetection experiments (Papanayotou et al., 2013)”.

2) A second point raised concerns the effects of the mutations on the level of cholesterol, which we were asked to address more clearly. We have measured the relative abundance of free cholesterol versus phosphatidylcholine (cholesterol/PC ratio) in control cells and TLCD1/2 siRNA-treated cells with or without palmitate challenge. No changes of note were detected and this information is provided in Figure 4—figure supplement 4D, where the y-axis has been more clearly labelled “cholesterol/PC ratio” rather than the previous “FC/PC ratio”. Note that the lipidomics provided in Figure 4—figure supplement 6 all come from the same experiments, which allows direct comparisons. In Author response image 1 we provide the results from two experiments in which the cholesterol/PC ratios were measured. No changes in cholesterol were detected in the second experiment. We therefore feel confident in our statement that differences in cholesterol content do not explain the effects of TLCD1/2 knockdown on membrane properties (subsection “Mammalian TLCD1 and TLCD2 are regulators of membrane composition and fluidity”, third paragraph).

Cholesterol levels were not measured in *C. elegans* since it is not a structural component of membranes in this organism. We now explicitly indicate this as follows: “cholesterol levels were not measured since this lipid does not appear to play a structural role in *C. elegans* but instead is a likely precursor for essential low-abundance metabolites (Merris et al., 2004).”

3) Also as requested in the cover letter, and by the reviewers, we have thoroughly revised the literature cited both in the Introduction and in the Discussion, and in particular raised the possibility that the FLD-1/TLCDs proteins may influence membrane composition by regulating the Lands cycle by which phospholipids can be remodelled (specific changes are detailed below). Several improvements to our descriptions of the data and statistics have also been made. The “n” for each method used is clearly stated in the Materials and methods, and we also added a statement about repeats of entire experiment (subsection “Statistics”).

If you can address all of the reviewers' points, and provide a point-by-point description of your revisions, then I would be happy to consult them again to ask if the paper is now of acceptable quality.Separate reviews (please respond to each point):

Reviewer #1:

[…] Specific points:1) Plasma membrane localization of FLD1. This localization is inferred from confocal images using a GFP construct (Figures 2F and Figure 1—figure supplement 4). I notice that the cells shown in Figure 2F have a prominent nucleus, making the cytoplasm volume minimal. Therefore, I'm not entirely convinced that the protein is at the plasma membrane. Wouldn't a protein present in the cortical ER give a similar localization pattern? This is important because, functionally, the localization of FLD1 at the plasma membrane is puzzling considering that lipid synthesis and remodeling is believed to occur mostly in internal membranes (notably ER). How can FLD1 affects fatty acid incorporation into phospholipids if the protein is not in the same membrane as lysophospholipid acyl transferases? See also my remark on the mammalian proteins.

Concerns plasma membrane localization of FLD-1 and the mammalian homologs. See answer to decision letter, point 1.

2) The genetic characterization in Figures 2 and 3 is extensive and compelling, showing a clear rescue effect of various fld-1 mutants on the defects induced by temperature or glucose on the growth and morphology of worm mutants for the PAQR-2 and/or IGLR-2 proteins. The genetic interactions tested in Figures 2D-F and 3E suggest that FLD-1 acts by a pathway different from the previously described pathway, which relies on the overexpression of a FA desaturase. Instead, they suggest a genetic interaction with the desaturases fat-6 or fat-7 (please comment on the lack of interaction with fat-5; Figure 3E). How about acyl transferases e.g. MBOA-6?

We now comment on the lack of interaction with *fat-5*, in Figure 3E. Specifically, we now write: “Interestingly, RNAi knockdown of the desaturases *fat-6* and/or *fat-7* abolished the suppression of *paqr-2(tm3410)* glucose intolerance by *fld-1(et48)*, and this was the case both in the presence or absence of the *mdt-15(et14) gof* allele (Figure 3E). Knockdown of the desaturase *fat-5* also abolished the suppression of *paqr-2(tm3410)* glucose intolerance by *fld-1(et48)* but not in the presence of *mdt-15(et14)* which likely drives sufficient compensatory induction of *fat-6* and *fat-7.*”

3) Related to the previous point. At least in text or perhaps through further genetic studies, the authors should try to connect their work with recent studies on the families of acyltransferases responsible for the incorporation of fatty acids with defined level of unsaturation (e.g. PMID:28578316 and others studies by the Shimizu group), as well as on recent attempts to better characterize the molecular properties of polyunsaturated phospholipids in membranes (e.g. Pinot et al., 2014 for endocytosis).

The reviewer asks that we discuss the possible involvement of acyltransferases as targets of FLD-1/TLCDs. We have now done so in the Discussion: “It is possible, for example, that FLD-1 and TLCD1/2 influence substrate selection by phospholipases or lysophospholipid acyltransferases, i.e. regulating the Lands cycle through which phospholipids are actively remodelled by fatty acid exchange and hence effectively influence membrane composition and properties (Pan and Storlien, 1993; Abbott et al., 2012; Andersson et al., 2002; Shindou et al., 2017). In particular, the presence of several membrane-bound acyl transferases in *C. elegans* suggest a promising avenue of genetic interaction studies to test their possible connection to FLD-1 activity (64,65).” Also, the work by Pinot et al. is now cited in the Introduction (Pinot et al., 2014).

4) The lipidomic analysis on the acyl chain profile performed on phosphatidylethanolamine is also compelling. Under glucose, the paqr-2 mutant accumulates saturated PE at the expense of polyunsaturated PE, an effect that is inverted by the further introduction of the fld-1 mutation, notably in combination with mutation of the mdt-15 gene. The data under conditions of a saturated diet (Figure 3—figure supplement 1B-D) are also impressive. The authors, however, might mention that the plasma membrane accounts for only a minor part of the total cellular membranes. Therefore, it is unlikely that the plasma membrane is the only one to be affected in its acyl chain profile. The lipidomic analysis is performed on cells or tissues, not on plasma membrane, which is technically much more demanding.

The reviewer brings up the fact that the phospholipid composition analyses presented in the manuscript likely reflect much more than just the plasma membrane. This is entirely correct and the reviewer’s comment alerted us to a possible lack of clarity on this point. We now write: “Note that the lipidomic analysis was performed on whole worms, and thus reflects global lipid composition rather than the composition of any specific membrane; also, cholesterol levels were not measured since this lipid does not appear to play a structural role in *C. elegans* but instead is a likely precursor for essential low-abundance metabolites (Merris et al., 2004).” Also, regarding human cells: “Note that, as in *C. elegans,* the lipidomic analysis was performed on whole cells, and thus reflects global lipid composition rather than the composition of any specific membrane.”

5) For mammalian cells, would it be possible to further study the sub-localization of the FLD-1 homologues (TLCD1 and TLCD2).

Concerns the localization of the TLCDs. See answer to decision letter, point 1.

Minor Comments:6) I am not sure I understood the experiments shown in Figure 4—figure supplement 2O and the associated conclusion. Is there an effect of silencing TLCD1 or TLCD2 on desaturases expression?

This comment concerns Figure 4—figure supplement 2O. There is indeed an effect of TLCD1/2 knockdown on SCD expression. We now clarify: “The expression levels of the desaturases SCD and FADS1-3 are also not increased by TLCD1 or TLCD2 knockdown (Figure 4—figure supplement 2O) ruling this out as a mechanism for the fluidizing effect; indeed, the SCD expression is downregulated in TLCD1 or TLCD2 knockdown which may reflect a decreased demand on their activity.”

Additional data files and statistical comments:The running title is: TLCD1/2 limit LCPUFAs. It should be TLCD1/2 limit LCPUFA incorporation.

Reviewer #2:

[…] Without any doubt, this manuscript identifies fld-1 and TLCD1/2 as important contributors to membrane homeostasis in C. elegans and human cells. The technical quality and presentation of the data is very good. The conclusions of the manuscript are justified and provide an important, new link for a better understanding of membrane lipids in health and disease. After a revision of the manuscript based on the points discussed below, I can recommend this manuscript for publication in eLife.Strength/Weaknesses:Even though the number of weaknesses outnumbers the number of strengths, I consider this manuscript as an excellent study with potentially very high impact on a much broader scientific community in the future.Strength 1: This manuscript represents a thorough genetic dissection of the pathways involved with membrane homeostasis paired with an extensive phenotypic characterization including lipidomic analyses.Strength 2: The technical quality of the data is very high and support the conclusions of the authors.Strength 3: The authors provide compelling and carefully controlled data from two distinct model systems: C. elegans and human cells.Weakness 1: The literature referenced in the Introduction and Discussion does often not suit the purpose. Many papers referred to in the text report correlative observations, but are cited in way to make a statement. Only few studies mentioned in the text provide the original, clear-cut, biochemical or biophysical studies for the point being made. In some instances, there seems to be a discrepancy between the cited reference and the statement made in the text. This must be improved.Examples:Weijers, 2016: The role of membrane 'flexibility' on microcirculation does not become clear to the reader. The term 'flexibility' lacks a clear physical definition.Weijers, 2012 is a review article with similar content as Weijers, 2016, while other highly influential references such as (e.g. PMID: 22385956) or original literature linking the membrane lipid composition to diabetes are missing.Taraboletti et al., 1989 and Sherbet, 1989 do not justify the statement that defects in membrane are hallmarks of cancer. Taraboletti et al., 1989 provides only a correlative argument, while the Abstract of Sherbet, 1989 even states 'Steady-state fluorescence polarization […] does not suggest marked differences in bulk 'fluidity' of metastatic variant'.Fecchio, Palazzi and de Laureto, 2018 primarily discusses the folding of α-synuclein in the presence of high concentration of free polyunsaturated fatty acids and in the absence of cellular membranes. This is not a suitable model to imply functions of PUFAs in neurodegenerative diseases. The role of α-synuclein in interacting with cellular membranes and regulating vesicular traffic is not being discussed at all in this context.Other recent studies, which suggest important roles in specific polyunsaturated lipids in membrane deformation (PMID:25104391), domain stability (PMID: 27119640), membrane fission events (PMID: 29543154). Potential players important for mediated the regulated production/conversion of membrane lipids in fld-1 cells are not discussed (PMID: 18782225; PMID: 18094042; PMID: 7174678). I suggest that the authors carefully revisit their list of references used in the Introduction and the Discussion.

The reviewer urged us to reconsider the literature cited in the Introduction and Discussion, and made several suggestions for improvement. The references in the first paragraph of the Introduction have been updated to include more primary research articles, and now include the reviewer’s suggestions which were all spot on. Similarly, the second paragraph of the Discussion now brings up phospholipases/acyltransferases (Lands cycle) as a possible regulatory point for FLD-1/TLCDs (see reviewer 1, point 3).

Weakness 2: The concept of membrane fluidity is highly prevalent in biology and serves as a wonderful descriptor of membrane biophysics, which is also intuitively understood by a broader readership. However, I do not find any clear evidence in this manuscript (nor in the cited references) that membrane fluidity is indeed the critical factor in the context of the observed phenotypes. The role of lipids containing polyunsaturated fatty acids has gained increasing attention in the past years (see below) and the authors could do a better job in highlighting these developments in light of their data.The authors should clearly indicate that membrane fluidity is used as a readily available descriptor of a membrane property. A separate paragraph should discuss that also other membrane properties are greatly affected by lipids with polyunsaturated fatty acid chains and that it is not clear, which of these membrane properties are most relevant for the observed phenotypes observed in genetic experiments.

At the reviewer’s suggestion, we now introduce “fluidity” as a convenient term alluding to a complex phenomenon: “Given its central and far-reaching importance, it is surprising that so little is known about the molecular mechanisms that regulate membrane composition and fluidity, a term used throughout this article as a proxy for membrane properties that include fluidity, phase behavior, thickness, curvature, intrinsic curvature and lateral pressure profile (Radanović et al., 2018; Mouritsen, 2011).” Additionally, we now include a new paragraph in the Discussion (third paragraph) that serves two purposes: mention the possible roles of LCPUFAs as precursors to signaling molecules, and reviews the evidence that *fld-1* actually acts through its effects on membrane properties.

Weakness 3: The important role of FLD-1 and TLCD1/2 is being established, but the potential targets of FLD-1 and TLCD1/2 are neither discussed, nor is there any speculation on how a signal from FLD-1 or TLCD1/2 might be transmitted to regulate the production/formation of very long chain polyunsaturated lipids. The authors should provide some more insight in how the communication between the plasma membrane bound proteins and their effector functions is being established. A direct regulation of FLD-1 by PAQR-2 is indicated in Figure 6, but I cannot find evidence that supports this speculation. Do the authors suggest a direct regulation or rather an indirect one via the fluidity (or other properties) of the plasma membrane?How can a plasma membrane embedded protein send a signal that affects the incorporation of polyunsaturated fatty acids into lipids? Does it signal go to the ER, to the nucleus, or to the plasma membrane? Is there any indication for a transcriptional response or for a signaling molecule/second messenger/post-translational modification that might transmit the signal to other proteins involved in establishing the fatty acyl chain composition of membrane lipids? Does, for example, knockdown of TLCD1/2 lower the activity of the unfolded protein response in response to palmitate? This would indicate that the lipid composition in the ER membrane is affected FLD-1 mediated regulation and not only the plasma membrane.

We completely agree with the reviewer. Our genetic studies clearly identify FLD-1/TLCDs as novel regulators of phospholipid composition acting on very long chain polyunsaturated fatty acid content. Nevertheless, these genetic studies have not revealed the actual molecular mechanism by which FLD-1/TLCDs regulate phospholipid composition. As mentioned earlier (reviewer 1, point 3 and reviewer 2, point 1) we now followed the suggestions of the reviewers and mention that FLD-1/TLCDs may influence the Lands cycle (Discussion, second paragraph). Also following up on the reviewer’s suggestion, we used a *hsp-4::GFP* reporter to investigate whether the ER-UPR is induced by exogenous SFAs in *paqr-2* mutants (achieved through glucose supplementation) and whether it is affected by the *fld-1* mutation (it is not). This is now included in Figure 1—figure supplement 6H. We did not investigate this further nor in mammalian cells. We are of course very curious about the molecular mechanism (signaling or otherwise) but we feel that this is beyond the scope of this study. Finally, regarding the possibility that *paqr-2* may regulate *fld-1*: this is presented as a dashed line in the model (Figure 6) and is mentioned only as a possibility consistent with the genetic interaction studies. This is now made clear in the legend: “The possibility that PAQR-2 regulates FLD-1 (dashed line) is suggested by the fact that loss-of-function mutations in *fld-1* suppress *paqr-2* mutant phenotypes; there is at present no evidence for physical interactions.” and in the text of the Discussion: “The suggestion that *paqr-2* may influence *tlcd-1/2* activity is raised only as a possibility consistent with the observed genetic interactions: we have at present no biochemical evidence that these proteins interact with each other nor do we know of a mechanism by which *paqr-2* could regulate *tlcd-1/2*.”

What is the connection to cholesterol?An important factor contributing to membrane fluidity and membrane phase behavior of cellular membranes in mammals is cholesterol. Especially because cholesterol has key functions in organizing the plasma membrane of mammalian cells (but not C. elegans), and because SREBP proteins are central regulators of many aspects of lipid metabolism, it would be important to see if and how TLCD1/2 affects the cholesterol level and cholesterol signaling in these cells.

The reviewer asks that we examine the effect of TLCD1/2 knockdown on cholesterol levels. See the response to decision letter, point 2 (above).

Related to the FRAP experimentsFor HEK cells and C. elegans different approaches have been undertaken. Does incubation at 37 °C of the BODIPY 500/510 for 10 min result in an uptake of the dye to inner membranes? In order words: do the FRAP data refer to FRAP in the plasma membrane or is it due to internal membranes? The authors should provide a confocal image that shows incorporation of the dye in the plasma membrane and no strong labeling of intracellular membranes. Furthermore, the authors should use a GPI-anchored protein to measure the effect of FLD-1 and/or TLCD1/2 on proteins on the extracellular leaflet of the plasma membrane.

The reviewer asks for clarifications about how the FRAP experiments are performed, especially regarding the use of BODIPY 500/510 C_1_, C_12_ to label membranes in mammalian cells. The reviewer is quite right: inner membranes are also labelled under our experimental conditions and are monitored in the FRAP experiments. We now provide an image to make that quite clear (Figure—figure supplement 2B and legend), as well as videos of three FRAP experiments (Figure 4—figure supplements 3-5: control siRNA, and TLCD2 siRNA with and without 400 μM palmitate).

Other points:General: The number of individual data points should be provided with the experiments and with the appropriate statistical tests.Subsection “fld-1 is ubiquitously expressed in plasma membranes”: The TLC (TRAM, LAG1,and CLN8) domain should be introduced better – otherwise it may remain unclear to many readers what TRAM, LAG1, and CLN8 stand for.

The TLCD domain is more clearly introduced (subsection “*fld-1* is ubiquitously expressed in plasma membranes”).

Subsection “fld-1 acts in a pathway distinct from other paqr-2 suppressors”: 'PA' is used as an abbreviation for palmitic acid, but introduced only later in the paragraph. Moreover, the abbreviation may be misleading, as the paper refers to phosphatidylethanolamine as PE, and PA might be misunderstood as phosphatidic acid. Maybe the authors find a less confusing way to refer to palmitic acid?

The abbreviation PA for palmitic acid is clarified at its first instance (subsection “*fld-1* acts in a pathway distinct from other *paqr-2* suppressors”).

Subsection “fld-1 acts in a pathway distinct from other paqr-2 suppressors”: the mdt-15 and cept-1 mutant worms are mentioned without further explanation of their putative functions. This might be confusing for a reader not familiar with C. elegans.

Information about *mdt-15* and *cept-1* is now provided (subsection “*fld-1* acts in a pathway distinct from other *paqr-2* suppressors”).

Subsection “Mammalian TLCD1 and TLCD2 are regulators of membrane composition and fluidity”, second paragraph and Figure 4—figure supplement 2O: The expression level of SCD is substantially reduced, but this observation is not discussed in the manuscript (only in the figure legend). Does this indicate that there are substantial changes in the lipid composition of the endoplasmic reticulum under these conditions, which lead to a distinct regulation of SCD?

Changes in SCD expression are now mentioned in the text (see reviewer 1, point 6).

The notion that 'as little as 1 µM of EPA or DHA […] prevents membrane rigidification by 400 µM PA' should be avoided. It does not make sense to compare these concentrations, because the uptake, metabolism, incorporation in different lipids, their subcellular localization, and turnover might differ substantially between these fatty acids. Just comparing the concentration in the media does not make sense.Subsection “Mammalian TLCD1 and TLCD2 are regulators of membrane composition and fluidity”, last paragraph: (as above) Comparing the concentration of LCPUFA necessary to protect cells from 200 µM PA is not relevant, given the complexity of the metabolic steps involved.Discussion, first paragraph: (similar as above): The expression 'fifty times more potent' is misleading and should be omitted. The cited reference (Yang et al., 2011) does not warrant equal metabolism of LCPUFAs and oleic acid in cells. This statement would ignore, that different fatty acids have remarkable distinct abundancies in the cellular membrane. The statement does not define in what respect LCPUFAs are more potent? How is it being measured? Of note, the effect of different lipids on collective bilayer properties are not linear. For these reasons, the authors should avoid a quantitative comparison of the effects by different fatty acids.

We have been much more cautious in describing the potency of LCPUFA supplements (subsection “Mammalian TLCD1 and TLCD2 are regulators of membrane composition and fluidity”, fourth paragraph; Discussion, first paragraph).

Discussion, second paragraph: The authors list a number of processes that seem to be affected by the membrane lipid composition. However, the true underlying mechanism remain entirely unexplored in these studies. Also, membrane microdomains are mentioned here for the first time in the manuscript. This of course has many implications on the interpretation of the FRAP experiments (see below). I would recommend not to go into a discussion on the modulatory role of the membrane composition, but instead focus more on the discussion how FLD-1 and/or TLCD1/2 might excerpt their modulatory function.

We removed mention of microdomains.

Discussion, third paragraph: It is unclear to me, how the presented data suggest a direct, regulatory role of PAQR-2 and IGLR-2 on FLD-1. The proteins might act independently. However, if PAQR-2 and IGLR-2 would modulate FLD-1 function – what would be the signal?

As mentioned earlier, the possibility of an interaction between PAQR-2 and FLD-1 is mentioned only as a possibility consistent with the genetics (see reviewer 2, point 3).

Figure 2C: The most interesting statistical test should be performed between worms with the fld-1 allele and without it.

The statistical test is done/shown for Figure 2C.

Additional data files and statistical comments:As mentioned in the previous comments, it would help if the number of experiments (n) would be provided with the experiments.

n for each experiment is stated in the Materials and methods section for each method. We also added a statement about repeats of entire experiment (subsection “Statistics”).

Reviewer #3:

We have the read the manuscript 'Membrane fluidity is regulated by the C. elegans transmembrane protein FLD-1 and its human homologs TLCD1/2' by Ruiz et al. In this study the authors have performed a suppressor screen for paqr (adiponectin receptor homolog) and iglr (adiponectin interacting protein) mutants that show increased membrane rigidity at low temperature or when grown in presented on high glucose. They identified several alleles of this gene were capable of suppressing the growth and morphological defects observed in the two mutants. They show that fld-1 mutants, under conditions of stress (cold/high glucose/or feeding of saturated fatty acid) continue to favor a membrane composition that is slightly higher in unsaturated fatty acids thus increasing its fluidity. They have demonstrated similar abilities for human homologs of this protein in tissue culture studies.Strengths of the study:1) Suppressor screen and identification of genetic interactors of PAQR/Adiponectin receptor.2) Detailed genetic studies of paqr/iglr/fld-1/and other previously isolated suppressors of paqr/iglr.3) Efforts to extend the original findings in C.elegans to human homologs in tissue culture experiments.4) Lipidomic correlation of morphological defects with increased saturated fatty acid content in membrane lipids (this is not new finding but extension of authors' findings in earlier studies).5) General concurrence between C. elegans and human data.Weakness of the study:1) No biochemical function for FLD-1/TLCD1-2 has been elucidated in this study. The manuscript begins as a genetic interaction study and remains as such despite valiant efforts by the authors to describe the interactions using numerous experiments and extending the studies to human homologs of FLD-1.

The reviewer raises the issue that we still do not know the biochemical function of FLD-1/TLCDs. This is true and was discussed in the answer to reviewer 2, point 3.

2) The Abstract is misleading –*'We isolated C. elegans mutants with improved tolerance to dietary saturated fat, including eight alleles of the novel gene fld-1 that encodes ahomolog of the human TLCD1 and TLCD2 transmembrane proteins. FLD-1 is localized on plasma membranes and mutants have an excess of highly membrane-fluidizing long-chain polyunsaturated fatty acid-containing phospholipids that results in improved membrane fluidity as measured* in vivo using fluorescent recovery after photobleaching (FRAP)'.The mutants do not show notable phenotypes or deviations from normal under regular growth conditions, at least from what can be inferred from data observed in the figures and by the authors' own admission while describing the phenotypic characterization of the mutants. The mutants do not show improved tolerance to saturated fat on their own and do not have an excess of membrane-fluidizing LCPUFA ordinarily but only under stress conditions. They show improved tolerance only in the presence of other mutations.

We have modified the Abstract to clarify that the beneficial effects of *fld-1* mutations are evident in sensitized genetic backgrounds: “We isolated *C. elegans* mutants that improved tolerance to dietary saturated fat in a sensitized genetic background, including eight alleles of the novel gene *fld-1* that encodes a homolog of the human TLCD1 and TLCD2 transmembrane proteins.”

3) Since the molecular function of FLD-1 has not been elucidated it leads to a conceptual dilemma in the working hypothesis proposed by the authors. The authors argue that FLD-1/TLCD1-2 is a plasma membrane protein that decreases the incorporation of LCPUFA to the membrane implying a function in the immediate vicinity of plasma membrane (thus cell autonomous). Yet in Figure 1—figure supplement 5C, they show transgenic expression of FLD-1 in hypodermis or intestine is sufficient to restore glucose sensitivity on paqr-fld-1 double mutant indicating a cell non-autonomous role.

We agree that the cell non-autonomous nature of the membrane homeostasis effect of *fld-1* may seem puzzling. We recently published an article showing that PAQR-2 and IGLR-2 can similarly maintain systemic membrane homeostasis in a cell nonautonomous manner, which is now mentioned: “This is consistent with an earlier study demonstrating that the maintenance of membrane homeostasis is cell nonautonomous and may rely on effective trafficking of lipids among tissues (Bodhicharla et al. 2018).”

4) The data shows that in the absence of these proteins, there is an increase in incorporation of LCPUFA's within 6 hours, indicating that the deacylation-reacylation cycle (Lands cycle) may be involved. Is it possible that FLD-1/TLCD1/TLCD2 interact with membrane phospholipases or Acyl-CoA:lysopospholipid acyltransferases and modulate their function?

Possible connections to the Lands cycle are now mentioned in the Discussion (second paragraph). See also answer to reviewer 1, point 3 and reviewer 2, point 3.

There are several minor corrections that need to be incorporated throughout the manuscript. A few examples are noted below:1) Figure 3E-G the dashed line needs to be included.

The dashed line has been added to Figure 3E (it is not needed in 3F-G).

2) Figure 4—figure supplement 3. Legend refers to '(R-S) Knockdown of the desaturase FADS2 reduces the membrane fluidity ofHEK293 cells cultivated in 400 μM PA but does not prevent the protective effect of TLCD2 knockdown'. These figures seem to be missing.

Legend to Figure 4—figure supplement 3 has been corrected.

3) Figure 2A-B: The fld-1(et45) shown as single mutant but it should be fld-1(et45); paqr-2 double mutant?

The labels in Figure 2A-B have been fixed.

4) Subsection “fld-1 acts in a pathway distinct from other paqr-2 suppressors”: typo error fld-1(et34) it should be fld-1(et48).

The typo has been fixed: it is *fld-1(et48)*.

5) It is not clear why SDs have been used in some analyses and SEM for others?

SEM is used throughout except when scoring the tail tip defects, which is a “Normal/Abnormal” scoring and for which the 95% confidence is indicated.